# TurtleAI: Benchmarking Visual Programming and Reasoning for Multimodal Models in Turtle Graphics

## Abstract

Multimodal vision-language models (VLMs) have achieved remarkable success in fundamental visual tasks like image captioning and visual question answering. However, their performance on complex visual tasks requiring integrated visual reasoning and problem-solving capabilities remains underexplored. To bridge this gap, we introduce TurtleAI, a multimodal benchmark to evaluate VLMs on visual programming and reasoning tasks in the Turtle Graphics domain. Our benchmark contains 823 visual programming tasks that challenge VLMs to generate Python code to replicate patterns in images. Evaluation of 20 VLMs reveals that state-of-the-art models like GPT-4o and Qwen2-VL-72B struggle with these tasks, achieving success rates of only 26.5% and 11.8% respectively. Our analysis reveals that models often fail to align their code implementation with visual reasoning. To address this misalignment, we propose TurtleAI-Datagen, a data generation framework that creates large-scale synthetic datasets consisting of task-code pairs. Using just 10 initial samples, TurtleAI-Datagen generates over 700k samples. Fine-tuning on this dataset significantly reduces errors arising from the misalignment between visual reasoning and program synthesis, improving Qwen2-VL-72B's performance by over 20%. We will release the benchmark publicly to facilitate future research.

## 1 Introduction

Large language models (LLMs) have demonstrated remarkable capabilities across various domains (Bubeck et al., 2023). Recent work has integrated visual modalities into LLMs, leading to the emergence of multimodal vision-language models (VLMs) like GPT-4o (OpenAI, 2024a) and Qwen2-VL (Wang et al., 2024). These VLMs extend their versatility to tasks requiring visual understanding and reasoning, such as image captioning (Ramesh et al., 2021), visual question answering (Radford et al., 2021; Yue et al., 2024), and visual math reasoning (Lu et al., 2024a), showcasing capabilities across diverse visual domains.

Despite these advancements, VLM performance on complex tasks requiring integrated visual reasoning and problem-solving capabilities remains underexplored. Real-world visual tasks often necessitate visual reasoning to understand, interpret, and analyze visual information, followed by problem-solving to devise effective solutions (Lu et al., 2024b; Badue et al., 2021; Billard & Kragic, 2019). For instance, in robotic manipulation, a robot needs to first visually comprehend the spatial relationships among objects and then plan a sequence of actions to manipulate the object to the desired location (Billard & Kragic, 2019). While some benchmarks focus on assessing visual reasoning capabilities (Hendrycks et al., 2021; Lu et al., 2024a), there remains a gap in evaluating how well VLMs can solve complex tasks that require integrated visual reasoning and problem-solving capabilities (e.g., programming).

To bridge this gap, we introduce TurtleAI, a novel benchmark for assessing VLMs' capabilities in visual programming and reasoning within the Turtle Graphics domain. The benchmark comprises 823 visual programming tasks, each requiring VLMs to generate Python code that reproduces a target image. Figure 1 illustrates example images and corresponding VLM outputs. To solve these tasks, a model needs to first understand visual patterns and reason about spatial relationships, such as layout, size, position, and angles. Then, it needs to translate this visual analysis into executable

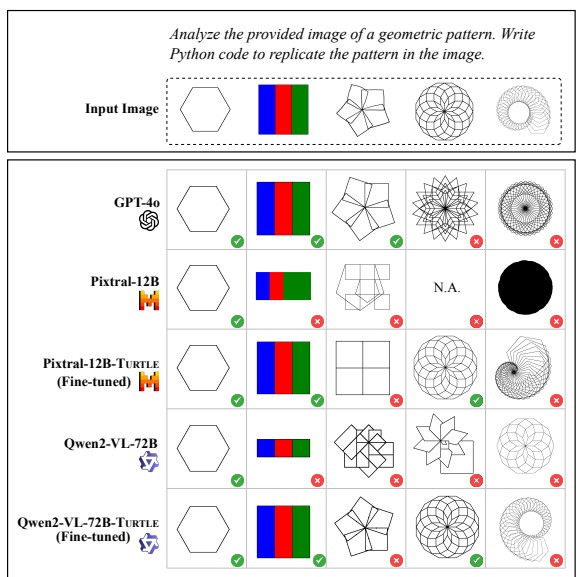

(a) VLMs' outputs for replicating different input images.

(b) Solution code for replicating the image ⬤.

Figure 1: Outputs of VLMs on visual-to-code generation tasks and an example solution code. (a) shows the input images and the visual outputs produced by executing each VLM's generated Python code, with success (✓) or failure (✗) shown for each output. (b) shows an example solution code for replicating the image ⬤.

Python code. For instance, drawing a flower-like dodecagon pattern (⬤) requires understanding the pattern, decomposing it into basic shapes (i.e., dodecagons), counting repetitions, reasoning about rotation angles, sequencing steps, and converting these reasoning outcomes into executable Python code (see Figure 1b). Successfully tackling these tasks requires integrated visual reasoning and problem-solving capabilities, presenting unique challenges for existing VLMs.

We evaluate various VLMs and find that state-of-the-art VLMs like GPT-4o (OpenAI, 2024a) and Qwen2-VL-72B (Wang et al., 2024) struggle with these tasks, achieving success rates of only 26.5% and 11.8% on basic programming tasks, respectively. We conduct systematic failure analysis and find that GPT-4o struggles most with spatial reasoning and accurate visual replication, while Qwen2-VL-72B struggles most with aligning the code implementation with visual reasoning. To address these limitations, particularly the code implementation alignment issues, we introduce TURTLEAI-Datagen, a novel data generation framework that uses large models to generate datasets consisting of task-code pairs. We leverage TURTLEAI-Datagen to use only 10 seed samples to generate over 700,000 synthetic task-code pairs. We fine-tune Qwen2-VL-72B on this synthetic dataset, achieving a 35.3% success rate, outperforming Qwen2-VL-72B by 23.5% and GPT-4o by 8.8%. Our analysis reveals that fine-tuning significantly reduces programming errors by 23.5%, leading to better alignment between program synthesis and visual reasoning.

Our contributions are as follows: First, we introduce TURTLEAI, a multimodal benchmark that evaluates VLMs' capabilities in program synthesis and visual reasoning, along with an automated evaluation framework for systematic evaluation. Second, we propose a novel data generation framework, TURTLEAI-Datagen, that can effectively generate large-scale synthetic datasets from a small set of seed samples. Third, we conduct comprehensive experiments and analyses on TURTLEAI, revealing valuable insights into VLMs' capabilities and limitations.

## 2 BACKGROUND AND SYNTHESIS OBJECTIVE

In this section, we provide background on Turtle Graphics and introduce the synthesis objective.

**Background on Turtle Graphics.** Turtle Graphics is a programmable method for creating vector graphics using a relative cursor (the "turtle") on a Cartesian plane (Python, 2024). Basic commands like "forward", "turn left", and "pen down" control the turtle's movement to draw lines and shapes.

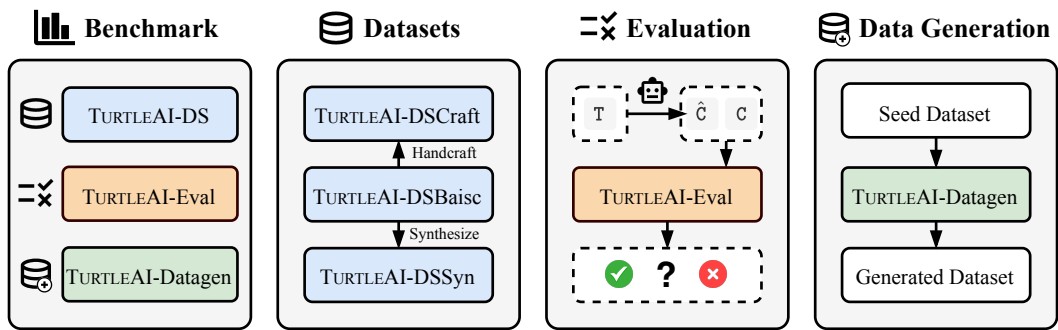

Figure 2: Overview of the TURTLEAI benchmark. TURTLEAI comprises three key components: (i) a collection of datasets TURTLEAI-DS for benchmarking, (ii) an evaluation framework TURTLEAI-Eval for assessing the correctness of generated code, and (iii) a data generation framework TURTLEAI-Datagen for generating synthetic datasets.

These commands can be combined with programming constructs such as loops, conditionals, and functions to generate visually appealing geometric patterns. Turtle Graphics is widely used in K-12 education to teach programming concepts and computational thinking (Staub, 2021; XLogoOnline, 2024; University of Oxford, 2025; CodeHS, 2025; Turtle Academy, 2025).

**Task specification.** We define a visual programming task in Turtle Graphics as $T := (img, ins)$, a tuple consisting of a target image $img$ and a text-based instruction $ins$. The target image specifies the desired visual output, while the instruction specifies the requirements for generating the code that will replicate the pattern shown in the target image.

**Code specification.** The code space for Turtle Graphics tasks is defined using the Python programming language. A *solution code* for a task $T$ is a Python code $C$ that, after being executed, can accurately replicate the target image $img$ and satisfy the requirements specified by the instruction $ins$. For consistent evaluation, the task's instruction requires the solution code to be synthesized as a function `draw(t)`, which takes a turtle object $t$ as input. For instance, Figure 1b shows a solution code that generates the image ◉.

**Program synthesis objective.** The synthesis objective is to develop a synthesizer function, $\mathcal{M} : T \rightarrow C$, which generates a solution code $C$ for a given task $T$ in Turtle Graphics. To evaluate $\mathcal{M}$ on a task $T$, we first use $\mathcal{M}$ to synthesize a code $\hat{C}$ which contains a Python function `draw(t)`. To evaluate the correctness of this Python function, a straightforward way is to compare the images generated by $\hat{C}$ and $C$ pixel-by-pixel. However, this pixel-wise comparison fails to account for differences in size, position, or line width of patterns being drawn in images (Marbach et al., 2022). In the next section, as part of our benchmark TURTLEAI, we will address this by introducing an evaluation framework.

## 3 THE TURTLEAI BENCHMARK

In this section, we first provide an overview of the benchmark, including its datasets, evaluation, and data generation, followed by a detailed description of each part.

### 3.1 OVERVIEW OF TURTLEAI

TURTLEAI is a benchmark within the Turtle Graphics domain. Figure 2 illustrates the key components of TURTLEAI, which consists of:

1. **TURTLEAI-DS**: a collection of evaluation datasets including (i) TURTLEAI-*DSBasic*, which is a basic dataset curated from the visual programming platform XLogoOnline (XLogoOnline, 2024), (ii) TURTLEAI-*DSCraft*, which is a hand-crafted dataset containing human-drawn tasks, and (iii) TURTLEAI-*DSSyn*, which is a synthetic dataset of selected high-quality synthesized tasks.

2. **TURTLEAI-Eval**: an evaluation framework for assessing the correctness of synthesized code.

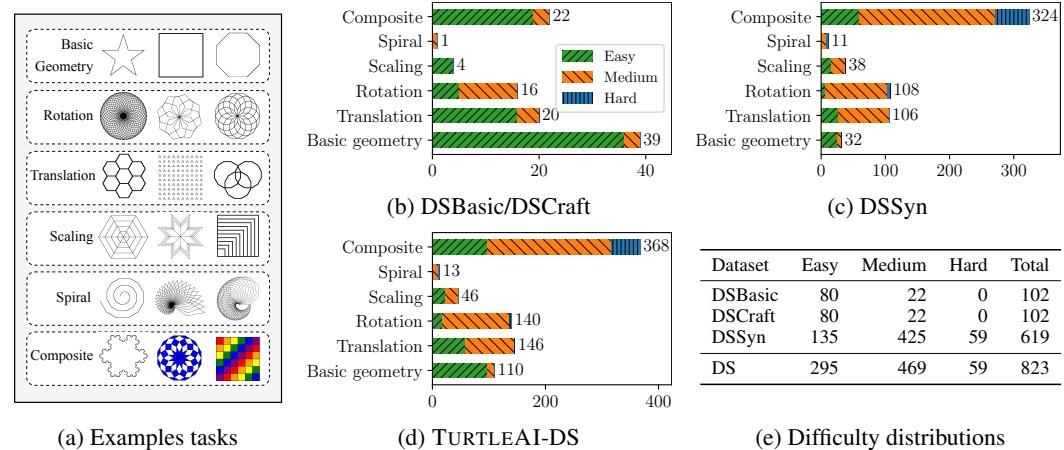

(a) Examples tasks    (b) DSBasic/DSCraft    (c) DSSyn    (d) TURTLEAI-DS    (e) Difficulty distributions

Figure 3: Dataset composition and statistics. Tasks are divided into six categories and three difficulty levels. (a) shows representative examples for each category. (b-d) show the distributions of these categories in TURTLEAI-DSBasic, TURTLEAI-DSSyn, and TURTLEAI-DS, respectively. Note that TURTLEAI-DSCraft has the same distribution as TURTLEAI-DSBasic. (e) shows the difficulty distribution across different datasets. Detailed labeling process for task categories and difficulty levels are provided in Appendix B.2.

3. **TURTLEAI-Datagen**: a data generation framework that synthesizes high-quality task-code pairs, used to generate both the evaluation dataset TURTLEAI-DSSyn and a large-scale training dataset TURTLEAI-Train.

## 3.2 EVALUATION DATASETS TURTLEAI-DS

TURTLEAI includes TURTLEAI-DS, a collection of 823 evaluation tasks organized into three distinct datasets. Figure 3 shows the example tasks and distribution of these datasets.

**TURTLEAI-DSBasic (Size 102).** This is a basic dataset containing tasks curated from the visual programming platform XLogoOnline (XLogoOnline, 2024). These tasks are originally designed by experts to teach the basic programming concepts of Turtle Graphics and have been widely used in the programming education domain (Staub, 2021).

**TURTLEAI-DSCraft (Size 102).** This is a hand-crafted dataset containing tasks generated by hand-drawing each task in TURTLEAI-DSBasic.

**TURTLEAI-DSSyn (Size 619).** This is a synthetic dataset containing tasks generated using TURTLEAI-DSBasic as a seed dataset for our data generation framework (see Section 3.4), followed by manual selection of high-quality tasks.

More details about the dataset generation process are provided in Appendix B.

## 3.3 EVALUATION FRAMEWORK TURTLEAI-EVAL

In our benchmark, evaluation checks whether a synthesized code $\hat{C}$ produces the semantically same drawing as the ground-truth code $C$. As discussed in Section 2, simple pixel-wise comparison fails to account for differences in size, position, or line width of the drawn pattern (Marbach et al., 2022). For example, a square with side length 100 would be considered different from a square with side length 101, even though they are semantically equivalent from a human perspective.

To address this issue, we propose a robust evaluation framework, TURTLEAI-Eval, which compares drawings in a transformed, normalized (canonical) space that is invariant to size, position, and line width. Our evaluation framework works as follows: First, we implement a customized Turtle Graphics emulator to execute $C$ and $\hat{C}$, generating their respective drawings. During execution, the emulator records all drawing states, such as coordinates and colors for drawing lines, filling polygon, etc. Second, we normalize the recorded coordinates to a fixed range, center the drawings at the origin, and standardize line widths to 1. Third, we sequentially render these normalized drawing states into

images, producing standardized images img and i$\hat{m}$g for C and Ĉ, respectively.[1] Finally, we provide the following two metrics to compare these two images:

- *Symbolic comparison*: This compares the standardized img and i$\hat{m}$g pixel-by-pixel. If the percentage of the same pixels between img and i$\hat{m}$g is above a predefined threshold, the comparison result is *success*; otherwise, the comparison result is *fail*.

- *Embedding-based comparison*: This compares the standardized img and i$\hat{m}$g from the embedding space. This is achieved by first extracting the image embeddings from img and i$\hat{m}$g using a pre-trained image encoder model such as ResNet18 (He et al., 2016). Then a similarity score is computed between these embeddings using a distance metric and normalized between range $[0, 1]$. If the similarity score exceeds a predefined threshold, the comparison result is *success*; otherwise, the comparison result is *fail*.[2]

The two comparison methods address different evaluation needs: symbolic comparison checks exact pixel-space equivalence, while embedding-based comparison measures semantic similarity, tolerating minor rotations and scaling that symbolic comparison rejects. We validate the accuracy of both methods against human evaluation, achieving 99.1% and 98.1% accuracy, respectively. Further analysis are provided in Appendix D.

### 3.4 DATA GENERATION FRAMEWORK TURTLEAI-DATAGEN

We evaluate various VLMs and find that state-of-the-art VLMs like GPT-4o (OpenAI, 2024a) and Qwen2-VL-72B (Wang et al., 2024) struggle with these tasks, achieving success rates of only 26.5% and 11.8% on TURTLEAI-DSBasic, respectively. To investigate the reasons, we analyze Qwen2-VL-72B's failures and find the most common error (30.4% of tasks) is failing to implement code consistent with its visual reasoning. This is likely due to insufficient training data on code-to-image alignment in Turtle Graphics. To address this, we propose TURTLEAI-Datagen, a novel framework that generates high-quality image-code pairs for supervised training to bridge the gap between visual reasoning and code implementation. TURTLEAI-Datagen is iterative, consisting of three stages described below.

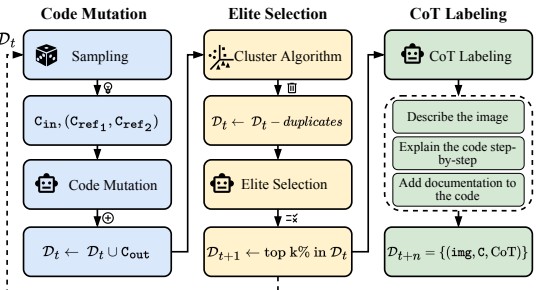

Figure 4: Illustration of the data generation framework TURTLEAI-Datagen, which comprises three stages: (i) code mutation to create diverse code variants via reference-guided mutation, (ii) elite selection to deduplicate and select high-quality samples, and (iii) CoT labeling to add CoT annotations.

**Stage 1: code mutation.** This stage aims to generate a larger set of codes from the seed dataset $\mathcal{D}_t$ (Xu et al., 2024; Ahmed et al., 2020; Wen et al., 2024). This can be done by pre-defining instructions for LLMs to mutate the codes (Xu et al., 2024). For example, one can use the instruction "add a loop to the code" to guide the LLM to mutate an input code $C_{in}$:

$$C_{out} = \text{LLM}(C_{in}, \text{instruction} = \text{``add a loop to the code''}). \tag{1}$$

However, a fixed set of pre-defined instructions limits the diversity of mutated codes by only capturing explicitly specified mutation patterns. To address this, we propose *reference-guided code mutation*, which uses a LLM to infer mutation patterns. In our approach, the LLM is given a pair of reference codes $(C_{ref_1}, C_{ref_2})$ and prompted to infer the high-level mutation pattern $m(C_{ref_1}, C_{ref_2})$. It then applies this pattern to another input code $C_{in}$, producing a new mutated output code:

$$C_{out} = \text{LLM}\big(C_{in}, \text{instruction} = m(C_{ref_1}, C_{ref_2})\big). \tag{2}$$

We apply this process to extend the seed dataset: for each $C_{in} \in \mathcal{D}_t$, we randomly sample $p$ reference code pairs from $\mathcal{D}_t$ and apply them to generate mutated codes. These codes are then executed to obtain images, producing a larger dataset of image-code pairs.

---

[1]For notational simplicity, we also use img to denote its standardized version when the context is clear.

[2]The threshold is chosen based on comparison with human-annotated ground truth (see Appendix D).

**Stage 2: elite selection.** In this stage, we first use a clustering algorithm to remove duplicated code-image pairs in the dataset. Then we use a VLM to score image quality based on predefined rubrics. Finally, we select the top $k\%$ image-code pairs, which serve as the seed dataset for the next iteration $t + 1$. After $n$ iterations, this produces a large-scale dataset $\mathcal{D}_{t+n}$.

**Stage 3: CoT labeling.** The final stage of TURTLEAI-Datagen labels each image-code pair in $\mathcal{D}_{t+n}$ with Chain-of-Thought (CoT) reasoning for step-by-step code generation (Wei et al., 2022; Zelikman et al., 2022). For each $(\texttt{img}, \texttt{C})$ pair, we instruct a VLM to generate CoT reasoning by first describing the image, then explaining the solution code step by step, and finally adding documentation. This produces a dataset of image-code pairs with CoT reasoning.

**Generating training dataset TURTLEAI-Train.** By leveraging TURTLEAI-Datagen, we generate a large-scale training dataset TURTLEAI-Train with $738,126$ image-code pairs. This is achieved by starting with a manually curated seed of 10 pairs and applying TURTLEAI-Datagen for 5 iterations. More details are provided in Appendix B.2.

## 4 EXPERIMENTAL EVALUATION

In this section, we evaluate VLMs on the TURTLEAI benchmark. We first describe the experimental setup in Section 4.1, then present the main results in Section 4.2, followed by failure analysis (Section 4.3), fine-tuning scaling analysis (Section 4.4), and out-of-distribution analysis (Section 4.5).

### 4.1 EXPERIMENTAL SETUP

**Benchmark datasets.** We use the four evaluation datasets described in Section 3.2 for evaluation, including TURTLEAI-DSBasic, TURTLEAI-DSCraft, TURTLEAI-DSSyn, and TURTLEAI-DS.

**Evaluation procedure and metrics.** Each evaluation dataset consists of $(\texttt{T}, \texttt{C})$ pairs. For each pair, we provide the task image $\texttt{img}$ and a fixed prompt instructing the model to generate the code snippet $\hat{\texttt{C}}$ in the desired format. The model may also output explanations, but we only extract $\hat{\texttt{C}}$. We then evaluate $\hat{\texttt{C}}$ against the ground-truth $\texttt{C}$ using our evaluation framework (Section 3.3). The *success rate* is the number of successful predictions divided by the total number of samples. We report both symbolic- and embedding-based success rates. For the main experiments, success rates are based on greedy decoding; Pass@K results from random sampling are provided in Appendix C.5.

**Models evaluated.** We compare various VLMs: (i) *Reasoning VLMs*, including GPT-family models: GPT-5 (OpenAI, 2025a), o3, and o4-mini (OpenAI, 2025b); (ii) *Non-reasoning Base VLMs*, covering model families from GPT (OpenAI, 2024a), Qwen (Wang et al., 2024), Molmo (Deitke et al., 2024), Llava (Li et al., 2024a), Pixtral (Agrawal et al., 2024), and InternVL (Chen et al., 2023); (iii) *Fine-tuned VLMs*, trained on our TURTLEAI-Train dataset with 738k samples, denoted with the TURTLE suffix. Full model details and fine-tuning are in Appendix E.2.

### 4.2 MAIN RESULTS

**TURTLEAI is challenging for all existing VLMs.** As shown in Figure 5, among base models, o3 achieves the highest symbolic success rate of $40.2\%$ on TURTLEAI-DSBasic, while o4-mini achieves the highest rate of $15.9\%$ on TURTLEAI-DS. Other open-source models perform worse, with success rates of $11.8\%$ and $6.6\%$ on TURTLEAI-DSBasic and TURTLEAI-DS, respectively.

**Fine-tuning helps yet remains limited.** Fine-tuned models Pixtral-12B-TURTLE, Qwen2-VL-7B-TURTLE, and Qwen2-VL-72B-TURTLE reach symbolic-based success rates of around $30\%$ on TURTLEAI-DSBasic and $15\%$ on TURTLEAI-DS, outperforming their base models by around $20\%$ and $10\%$, respectively. This shows that TURTLEAI-Datagen can generate large-scale synthetic datasets that effectively boost VLM performance on our benchmark. However, given the inherent challenges of TURTLEAI, fine-tuned performance remains far from satisfactory.

### 4.3 FAILURE ANALYSIS

To examine VLM limitations, we conduct a systematic failure analysis on three representative models: GPT-4o, Qwen2-VL-72B, and Qwen2-VL-72B-TURTLE, using the TURTLEAI-DSBasic

| | Size | TURTLEAI-DSBasic | | TURTLEAI-DSCraft | | TURTLEAI-DSSyn | | TURTLEAI-DS | |
|---|---|---|---|---|---|---|---|---|---|
| | | Sym. (%) | Emb. (%) | Sym. (%) | Emb. (%) | Sym. (%) | Emb. (%) | Sym. (%) | Emb. (%) |
| *Reasoning:* | | | | | | | | | |
| o3 | - | **40.20** | **44.12** | 8.82 | 9.80 | 10.18 | 9.21 | 13.73 | 13.61 |
| o4-mini | - | 36.27 | 38.24 | **28.43** | **29.41** | **10.50** | **10.50** | **15.92** | **16.28** |
| GPT-5 (medium) | - | 27.45 | 29.41 | 0.98 | 0.98 | 7.43 | 6.79 | 9.11 | 8.87 |
| *Non-reasoning (≥ 72B):* | | | | | | | | | |
| GPT-4o | - | 26.47 | 28.43 | 12.75 | 13.73 | 5.82 | 5.17 | 9.23 | 9.11 |
| GPT-4V | - | 15.69 | 17.65 | 8.82 | 10.78 | 3.23 | 2.75 | 5.47 | 5.59 |
| Pixtral-Large | 124B | 10.78 | 11.76 | **13.73** | **15.69** | 4.68 | 3.88 | 6.56 | 6.32 |
| Qwen2-VL | 72B | 11.76 | 14.71 | 7.84 | 8.82 | 1.45 | 1.62 | 3.52 | 4.13 |
| Llava-OneVision | 72B | 4.90 | 3.92 | 6.86 | 5.88 | 0.97 | 0.97 | 2.19 | 1.94 |
| InternVL2 | 76B | 11.76 | 13.73 | 7.84 | 8.82 | 1.13 | 0.65 | 3.28 | 3.28 |
| Molmo | 72B | 3.92 | 4.90 | 4.90 | 4.90 | 1.62 | 1.45 | 2.31 | 2.31 |
| NVLM-1.0-D | 72B | 0.00 | 0.00 | 0.00 | 0.00 | 0.16 | 0.16 | 0.12 | 0.12 |
| Qwen2-VL-TURTLE | 72B | **35.29** | **39.22** | 6.86 | 6.86 | **19.06** | **17.12** | **19.56** | **18.59** |
| *Non-reasoning (≤ 12B):* | | | | | | | | | |
| Pixtral | 12B | 9.80 | 9.80 | 2.94 | 2.94 | 0.97 | 0.97 | 2.31 | 2.31 |
| Qwen2-VL | 7B | 0.98 | 0.98 | 0.00 | 0.00 | 0.00 | 0.00 | 0.12 | 0.12 |
| Llava-OneVision | 7B | 3.92 | 3.92 | 2.94 | 2.94 | 0.32 | 0.48 | 1.09 | 1.22 |
| InternVL2 | 8B | 0.00 | 0.98 | 0.98 | 1.96 | 0.00 | 0.00 | 0.12 | 0.36 |
| GLM-4V | 9B | 0.00 | 0.00 | 0.00 | 0.98 | 0.00 | 0.00 | 0.00 | 0.12 |
| Molmo | 7B | 0.00 | 0.00 | 0.00 | 0.00 | 0.00 | 0.00 | 0.00 | 0.00 |
| Pixtral-TURTLE | 12B | 27.45 | 29.41 | **9.80** | **9.80** | **13.41** | **12.12** | **14.70** | **13.97** |
| Qwen2-VL-TURTLE | 7B | **28.43** | **30.39** | 6.86 | 8.82 | 11.95 | 11.15 | 13.37 | 13.24 |

Figure 5: Performance comparison of VLMs on different datasets. We evaluate VLMs using both symbolic comparison (Sym.) and embedding-based comparison (Emb.), with results shown as success rates (%). Fine-tuned models are denoted by the suffix TURTLE. The best performance within each group is shown in **bold**, and the second-best is underlined.

dataset. We manually review generated images, code, and available explanations to identify root causes of errors, attributing each case to the failure type that contributes most.[3] The distribution of failure types is shown in Figure 6, with definitions and examples in Appendix B.3.

**Models consistently struggle with spatial reasoning.** We find that all models struggle with spatial reasoning, which is the ability to reason about the spatial relationships between different patterns in the image, such as relative positions, distances, angles, and sizes of patterns. This might be due to the scarcity of training data that captures spatial relationships when training VLMs.

**Models struggle with precise visual details.** We find that despite correct visual reasoning, models still face difficulties in achieving visual precision during replication. For instance, both GPT-4o and Qwen2-VL-72B can often replicate the intended image from a high-level perspective but fail to achieve low-level visual accuracy, such as ignoring tiny details like angles and relative positions.

**Models often miss crucial details in images.** Visual understanding errors remain common between Qwen2-VL-72B and Qwen2-VL-72B-TURTLE. During review, we found that they often overlook small but crucial details and describe images using approximate common patterns. For instance, if an image shows a square with a unique cut-off, the models might just describe and draw a regular square, ignoring the specific cut-off.

**Fine-tuning improves code-reasoning alignment.** By comparing Qwen2-VL-72B and Qwen2-VL-72B-TURTLE, we observe that fine-tuning increases the success rate from $10.8\%$ to $35.3\%$, mainly by reducing programming errors (from $30.4\%$ to $6.9\%$). Visual understanding and spatial reasoning errors remain largely unchanged, suggesting that fine-tuning primarily improves alignment of code with visual reasoning rather than the reasoning itself.

## 4.4 SCALING OF FINE-TUNING PERFORMANCE WITH DATASET SIZE

We study how fine-tuning performance scales with the size of datasets generated by TURTLEAI-Datagen across different iterations. To this end, we fine-tune Pixtral-12B on datasets from each iteration and evaluate the resulting models, as shown in Figure 6d. The dataset grows exponentially with the number of iterations, at a rate of roughly 9.7, meaning each iteration produces a dataset approximately 9.7 times larger than the previous one. Performance improvements depend on the

---

[3]For failure analysis, we apply CoT prompting to Qwen2-VL-72B to elicit image descriptions and reasoning, yielding a $10.78\%$ success rate on TURTLEAI-DSBasic, close to non-CoT's $11.76\%$.

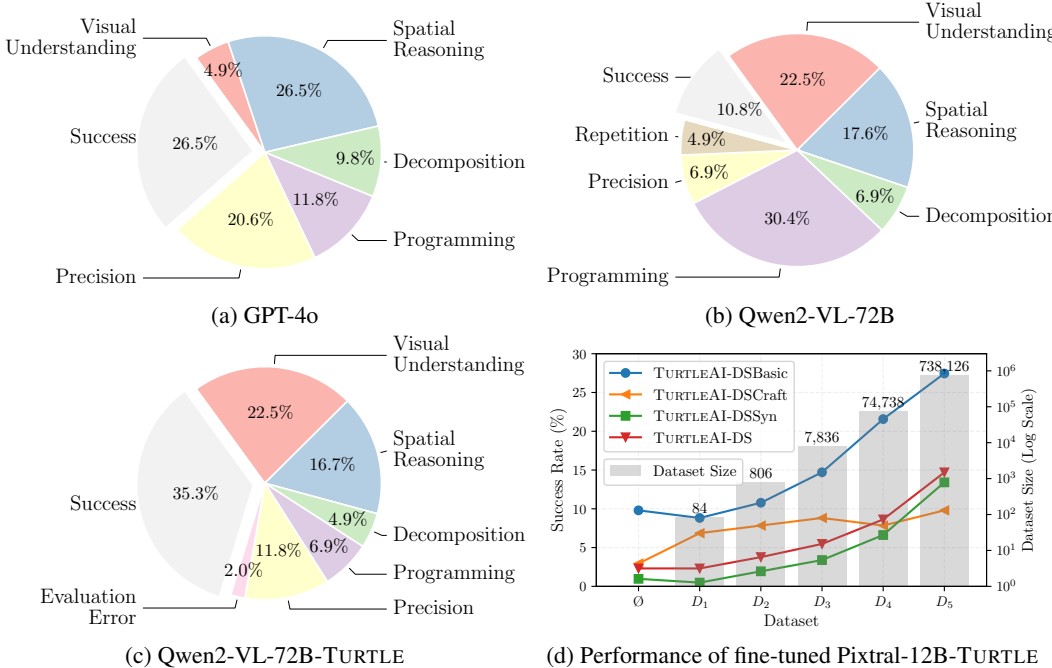

(a) GPT-4o

(b) Qwen2-VL-72B

(c) Qwen2-VL-72B-TURTLE

(d) Performance of fine-tuned Pixtral-12B-TURTLE

Figure 6: Analysis of failure types and scaling of fine-tuning performance. (a-c) show the distribution of failure types for three VLMs on TURTLEAI-DSBasic. (d) shows the performance of fine-tuned Pixtral-12B-TURTLE using datasets generated across different iterations.

| | DSBasic | DSCraft |
|---|---|---|
| Qwen2-VL-72B | 11.76 | **7.84** |
| Qwen2-VL-72B-TURTLE | **35.29** | 6.86 |
| Qwen2-VL-7B | 0.98 | 0.00 |
| Qwen2-VL-7B-TURTLE | **28.43** | **6.86** |
| Pixtral-12B | 9.80 | 2.94 |
| Pixtral-12B-TURTLE (w/ CoT) | **27.45** | **9.80** |
| Pixtral-12B-TURTLE (w/o CoT) | 22.55 | 0.00 |

| | HumanEval+ | MBPP+ |
|---|---|---|
| Qwen2-VL-72B | **85.4** | **77.5** |
| Qwen2-VL-72B-TURTLE | 67.7 | 73.5 |
| Qwen2-VL-7B | **70.7** | **55.3** |
| Qwen2-VL-7B-TURTLE | 28.0 | 33.1 |

(a) Success rates (%) on in-domain datasets.

(b) Success rates (%) on out-of-domain datasets.

Figure 7: (a) Success rates (%) on in-domain datasets; DSCraft contains hand-drawn OOD tasks from the same domain. (b) Pass@1 success rates (%) on out-of-domain program synthesis benchmarks; fine-tuned models are not tuned for these tasks.

dataset: for TURTLEAI-DSBasic, fine-tuning scales linearly with dataset size, while for TURTLEAI-DSSyn, the improvement is nearly exponential. In contrast, for the out-of-distribution dataset TURTLEAI-DSCraft, performance saturates after the first iteration and remains stable in subsequent iterations. These results suggest that exponentially larger datasets generated by TURTLEAI-Datagen can improve fine-tuning performance, although out-of-distribution datasets like TURTLEAI-DSCraft do not benefit from the increased data.

## 4.5 OUT-OF-DISTRIBUTION ANALYSIS OF FINE-TUNED MODELS

We analyze fine-tuned models on both in-domain and out-of-domain out-of-distribution (OOD) tasks, providing insights into their generalization capabilities. Results are shown in Figure 7.

**Fine-tuning can preserve or improve in-domain OOD capability via CoT.** We examine whether fine-tuned models can solve in-domain out-of-distribution tasks by comparing them with their base models on DSBasic and the hand-drawn OOD dataset DSCraft. DSCraft is created by hand-drawing images from DSBasic tasks and thus serves as an in-domain OOD dataset. As shown in Figure 7a, fine-tuning consistently improves performance on DSBasic, but offers little benefit on DSCraft for moderately performing models (Qwen2-VL-72B) and noticeable gains for weaker ones (Qwen2-VL-7B, Pixtral-12B). We hypothesize that CoT labeling stage in the data generation process may enhance

OOD performance, as it generates image descriptions that help ignore irrelevant variations in hand-drawn images. To test, we ablate CoT labeling stage in the data generation and fine-tune Pixtral-12B to obtain Pixtral-12B-TURTLE (w/o CoT). As shown in Figure 7a, this model fails on DSCraft despite outperforming Pixtral-12B on DSBasic, showing that CoT labeling is useful for OOD generalization.

**Fine-tuning leads to forgetting in out-of-domain tasks.** We investigate whether fine-tuning affects performance on out-of-domain tasks. To this end, we test fine-tuned Qwen2-VL models against their base versions on out-of-domain benchmarks, including HumanEval+ and MBPP+ (Liu et al., 2023). Pass@1 success rates are shown in Figure 7b. The results show a clear drop in performance after fine-tuning, consistent with prior work showing that fine-tuning may lead to some degrees of forgetting (Zeng et al., 2024a; Li et al., 2024b). This forgetting issue is more pronounced in the 7B model, suggesting smaller models are more prone to overfitting and losing general capabilities. The 72B model retains better generality, which might due to the higher capacity in retaining general knowledge.

## 5  RELATED WORK

**Program synthesis benchmarks.** Program synthesis aims to generate programs from specifications (Gulwani et al., 2017). Most benchmarks evaluate LLMs on generating code from natural language or docstrings (Chen et al., 2021; Rozière et al., 2023; Fried et al., 2023; Nijkamp et al., 2023; Austin et al., 2021). Recent work has extended this to visual programming, where code is generated from visual input (Wen et al., 2025; Gupta & Kembhavi, 2023; Padurean & Singla, 2024), though typically limited to domain-specific programs with simple code spaces. Our benchmark instead targets visual program synthesis in Python, emphasizing semantically richer and more complex code structures.

**Program synthesis for inverse graphics.** This task involves generating programs that reconstruct given graphical images. Prior work has studied it across domains such as SVG (Rodriguez et al., 2025; Zou et al., 2024), scientific figures (Belouadi et al., 2024), and Turtle Graphics (Ellis et al., 2021; Li & Ellis, 2024). We benchmark VLMs in Turtle Graphics, the vector graphics widely used in programming education (XLogoOnline, 2024; University of Oxford, 2025). Earlier work provided a preliminary evaluation of 3 VLMs on 260 tasks (Rismanchian et al., 2024), whereas our benchmark delivers a larger-scale, more comprehensive study, covering 20 models on 800+ tasks.

**Synthetic dataset generation using large models.** Large models are widely used to generate synthetic datasets across domains, including instruction-following, code, and CoT reasoning (Wang et al., 2023; Xu et al., 2024; Wei et al., 2024; Zelikman et al., 2022; Haluptzok et al., 2023). Such generation typically begins with a seed dataset and follows either a specification- or sample-based approach. The former relies on human-curated specifications to guide LLMs (Xu et al., 2024; Luo et al., 2024), while the latter draws from the seed dataset to provide context or inspiration (Haluptzok et al., 2023; Wang et al., 2023; Wei et al., 2024). Our framework adopts the sample-based approach but leverages samples as references to infer mutation patterns.

## 6  CONCLUDING DISCUSSIONS

**Summary.** In this paper, we introduced TURTLEAI, a multimodal benchmark for evaluating VLMs on visual programming and reasoning tasks. The benchmark comprises 823 tasks, each requiring the model to generate Python code to replicate given geometric patterns. Our evaluation revealed that existing VLMs struggle significantly in this domain, particularly in aligning program synthesis with visual reasoning. To address this challenge, we proposed TURTLEAI-Datagen, a novel data generation framework that generates high-quality synthetic data for alignment training. By leveraging this approach to create a 738k dataset and fine-tuning Qwen2-VL-72B, we achieved state-of-the-art performance with a 35.3% success rate while reducing code-reasoning alignment errors by over 20%.

**Limitations and future work.** We discuss some limitations of our work and propose ideas for addressing them in the future. First, our evaluation framework includes a normalization step that makes drawing comparison invariant to size, position, and line width, which may discard meaningful geometric variations that are essential in inverse graphics tasks. Future work could explore evaluation methods that preserve these geometric properties. Second, fine-tuned models struggle with out-of-distribution tasks. Future work could explore techniques such as random noise injection and domain mixup to improve generalization.

REPRODUCIBILITY STATEMENT

For better reproducibility, we provide some references that help with reproducing the results. Specifically, we provide the source code in the supplementary material, including the datasets, dataset generation scripts, and model evaluation artifacts. In Appendix B, we provide detailed descriptions of the dataset generation and labeling process. Appendix E contains implementation details for the data generation, model fine-tuning, model evaluation, and the evaluation framework. The prompts used in this paper are provided in Appendix G.

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

APPENDIX

**Table of Contents**

## A BROADER IMPACTS AND DECLARATION OF LLM USAGE

### A.1 BROADER IMPACTS

This work introduces a benchmark for evaluating the performance of existing vision language models (VLMs) in solving complex visual programming and reasoning tasks in the Turtle Graphics domain. In addition, we propose a novel data generation framework, TURTLEAI-Datagen, for generating large-scale synthetic data to train VLMs.

Our benchmark and data generation framework have several positive broader impacts: (i) our work can facilitate programming education, especially in K-12 settings where turtle graphics is commonly used, by enabling enhanced VLMs that power educational tools with real-time hints and feedback; (ii) our work can help track and improve VLMs' ability to understand and generate turtle graphics code, making it easier for both beginners and experts to create vector graphics and complex geometric art; and (iii) our data generation framework can advance synthetic data generation in fields where real data is scarce or difficult to collect, potentially helping researchers and developers build better models in a variety of domains beyond turtle graphics.

However, it is essential to acknowledge the potential risks associated with synthetic data generation. For instance, our framework could be misused to generate images with political or sensitive content. We emphasize the need for careful oversight and ethical considerations in the application of our framework to ensure that it is used responsibly and for the benefit of society.

### A.2 DECLARATION OF LLM USAGE

We declare that large language models (LLMs) were used only to assist with writing and formatting the manuscript. LLMs were not involved in the design of experiments, analysis of results, or any other important aspects in this paper.

# B  ADDITIONAL DETAILS ABOUT THE DATASET GENERATION AND LABELING PROCESS

In this section, we provide detailed information and generation processes about datasets used in this paper. After that, we describe our labeling process for image categories, difficulty levels, and failure types.

| Dataset | # Samples | Purpose | Seed Dataset | Seed Size |
|---|---|---|---|---|
| TURTLEAI-DS | 823 | Evaluation | - | - |
| TURTLEAI-DSBasic | 102 | Evaluation | - | - |
| TURTLEAI-DSCraft | 102 | Evaluation | TURTLEAI-DSBasic | 102 |
| TURTLEAI-DSSyn | 619 | Evaluation | TURTLEAI-DSBasic | 102 |
| TURTLEAI-Train | 738,126 | Train & Validation | Manually curated | 10 |

Figure 8: A summary of the datasets used in this paper.

## B.1  DATASET LICENSE

The TURTLEAI-DSBasic dataset is collected from the visual programming platform XLogoOnline and is licensed under CC BY-NC 4.0.[4].

The datasets TURTLEAI-Train and TURTLEAI-DSSyn were generated with vision–language models. The used models and their licenses are listed below:

- *Qwen2-VL-72B-Instruct* uses Qwen License Agreement.[5]

- *Pixtral-Large* uses Mistral Research License (MRL) for research/educational use.[6]

- *Llama 3.1-70B-Instruct* uses Llama 3.1 Community License Agreement.[7]

## B.2  DATASET GENERATION PROCESS

In this subsection, we provide more details about the generation process of different datasets used in this paper, including the evaluation datasets TURTLEAI-DSBasic, TURTLEAI-DSCraft, and TURTLEAI-DSSyn, and the training dataset TURTLEAI-Train. Figure 8 gives a summary of the datasets used in this paper.

**TURTLEAI-DSBasic.** This dataset is curated from the visual programming platform XLogoOnline. The tasks in this platform are carefully designed by domain experts has been used by tens of thousands of students for learning programming every year (Staub, 2021). The involvement of domain experts ensures that this dataset includes a diverse range of high-quality tasks, reflecting real-world learning scenarios.

**TURTLEAI-DSCraft.**

This dataset is generated by manually drawing the task images from TURTLEAI-DSBasic using a drawing tool. Specifically, we use each task image in TURTLEAI-DSBasic as the reference image and ask a human without any prior knowledge of Turtle Graphics or professional drawing skills to manually draw the task image using a digital drawing tool (i.e., an iPad). Finally, we replace each task image in TURTLEAI-DSBasic with the corresponding hand-drawn image, resulting in the dataset TURTLEAI-DSCraft. Figure 9 shows some examples of the reference images and the corresponding hand-drawn images in the dataset TURTLEAI-DSCraft. This dataset can be used to evaluate the generalization capabilities of the model to real-world drawing tasks.

---

[4]https://xlogo.inf.ethz.ch/terms.html
[5]https://huggingface.co/Qwen/Qwen2-VL-72B-Instruct/blob/main/LICENSE
[6]https://mistral.ai/news/pixtral-large
[7]https://github.com/meta-llama/llama-models/blob/main/models/llama3_1/LICENSE

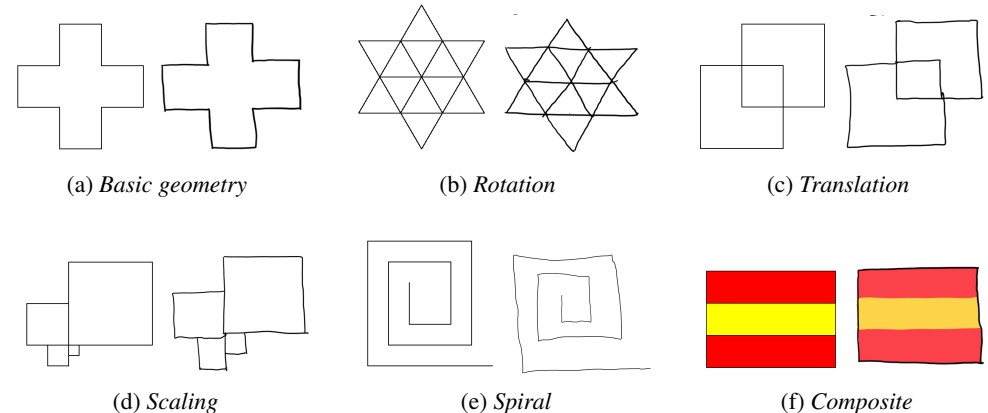

(a) *Basic geometry*     (b) *Rotation*     (c) *Translation*

(d) *Scaling*     (e) *Spiral*     (f) *Composite*

Figure 9: Examples of the reference images and the corresponding hand-drawn images in the dataset TURTLEAI-DSCraft. The reference images are shown on the left, and the corresponding hand-drawn images are shown on the right. One example is shown for each task category.

**TURTLEAI-DSSyn.** The dataset TURTLEAI-DSSyn is generated by our proposed data synthesis framework TURTLEAI-Datagen. Specifically, we use TURTLEAI-DSBasic (Size 102) as the seed dataset for TURTLEAI-Datagen. We iterate our TURTLEAI-Datagen over 3 iterations, each iteration we keep the top 30% of the generated samples for the next iteration of TURTLEAI-Datagen, resulting in a synthetic dataset with 8,214 image-code pairs. To further ensure the quality, we manually select from this dataset using the rubrics defined in the elite selection stage and make a binary decision to decide whether to keep an image-code pair. When making the decision, we adopt the rubrics used in the elite selection stage of TURTLEAI-Datagen, including (i) geometric structure and symmetry, (ii) visual appeal, clarity, and simplicity, (iii) structural coherence, (iv) alignment and positioning, (v) educational value and solvability, and (vi) color usage and necessity. When evaluating the quality of each sample, we make a binary decision (i.e., "good" or "bad") for each dimension, and only keep the sample if all dimensions are evaluated as "good". After this process, we obtain the final dataset TURTLEAI-DSSyn with 619 high-quality samples. Note that we don't apply the CoT labeling stage for generating TURTLEAI-DSSyn since this dataset is not used for training. The implementation details of TURTLEAI-Datagen are provided in Appendix E.1.

**TURTLEAI-DS.** The dataset TURTLEAI-DS is a union of TURTLEAI-DSBasic, TURTLEAI-DSCraft, and TURTLEAI-DSSyn datasets, including a total of $102 + 102 + 619 = 823$ samples.

**TURTLEAI-Train.** The dataset TURTLEAI-Train is a large-scale training dataset containing 738,126 samples. This dataset is generated by using only a seed dataset with only 10 seed examples. These seed examples are provided in Figure 10a. These seed examples are selected based on the following three principles:

- Minimal manual effort: The set should be as small as possible to reduce manual effort.
- Simplicity: The pairs should be easy to design and understand.
- Conceptual diversity: The set should cover a broad range of geometric transformation concepts.

By following these principles, we arrived at a set of 10 pairs that is both minimal and simple, while remaining diverse. These pairs capture a range of geometric transformation types observed in our domain, including:

- Adding or removing edges (e.g., transforming a triangle into a square or vice versa)
- Rotating shapes (e.g., turning a square into a diamond)
- Translating shapes (e.g., placing two squares side by side)
- Scaling (e.g., comparing a large square with a smaller one)
- Changing edge color (e.g., a square with red edges)

- Changing fill color (e.g., a red-filled square)
- Combining shapes (e.g., combining a square and a triangle)

After preparing for the seed examples, we generate the training dataset using the same settings as generating TURTLEAI-DSSyn, except that (i) we use a different seed dataset with only 10 manually designed examples; (ii) we iterate 5 times over the TURTLEAI-Datagen; (iii) we use $k = 70\%$ for the elite selection stage; (ii) we apply the CoT labeling after iterating 5 times. Examples of generated sampled and the generated CoT label are shown in Figure 10.

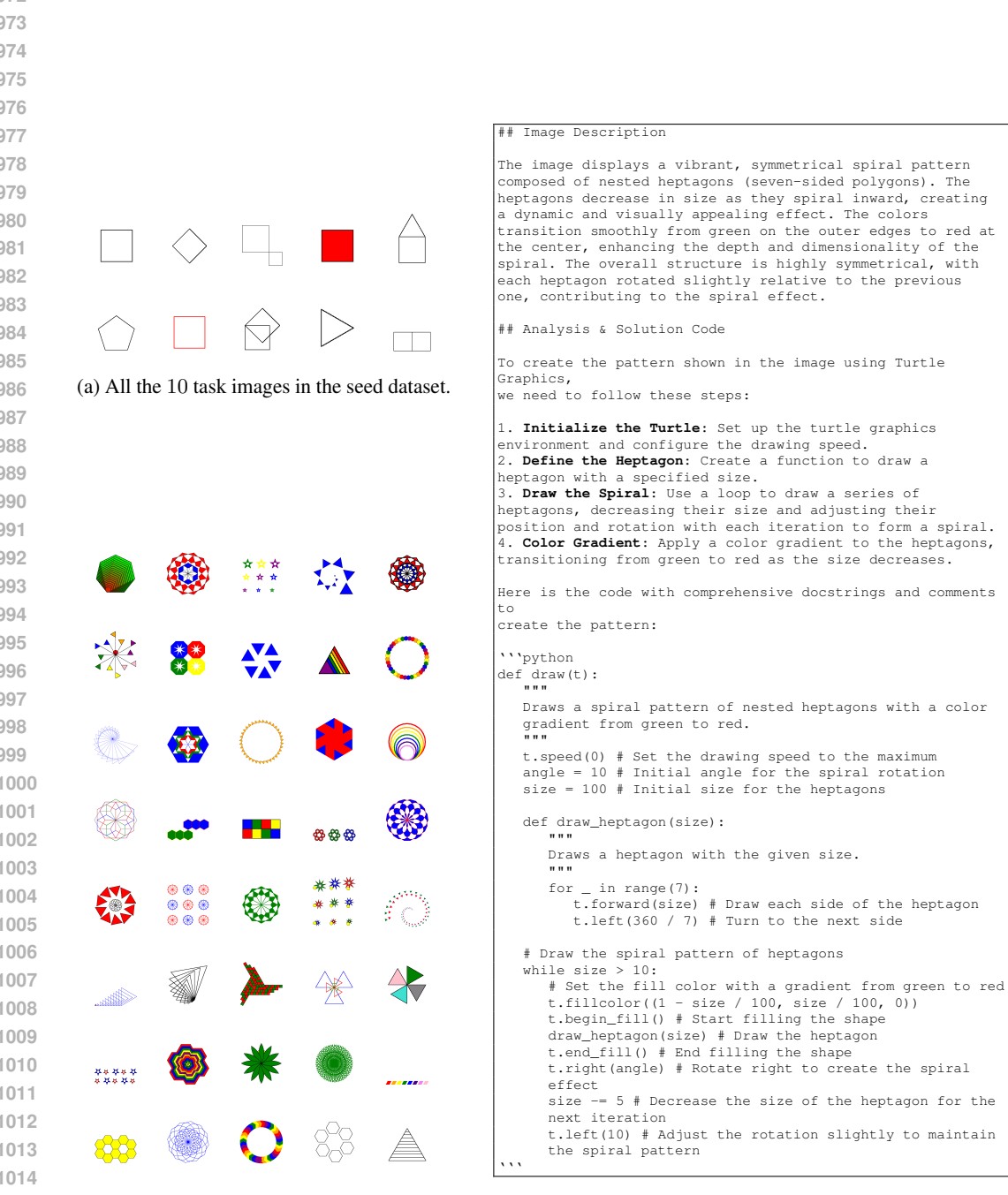

(a) All the 10 task images in the seed dataset.

(b) Generated images in TURTLEAI-Train.

## Image Description

The image displays a vibrant, symmetrical spiral pattern composed of nested heptagons (seven-sided polygons). The heptagons decrease in size as they spiral inward, creating a dynamic and visually appealing effect. The colors transition smoothly from green on the outer edges to red at the center, enhancing the depth and dimensionality of the spiral. The overall structure is highly symmetrical, with each heptagon rotated slightly relative to the previous one, contributing to the spiral effect.

## Analysis & Solution Code

To create the pattern shown in the image using Turtle Graphics,
we need to follow these steps:

1. **Initialize the Turtle**: Set up the turtle graphics environment and configure the drawing speed.
2. **Define the Heptagon**: Create a function to draw a heptagon with a specified size.
3. **Draw the Spiral**: Use a loop to draw a series of heptagons, decreasing their size and adjusting their position and rotation with each iteration to form a spiral.
4. **Color Gradient**: Apply a color gradient to the heptagons, transitioning from green to red as the size decreases.

Here is the code with comprehensive docstrings and comments to
create the pattern:

```python
def draw(t):
    """
    Draws a spiral pattern of nested heptagons with a color
    gradient from green to red.
    """
    t.speed(0) # Set the drawing speed to the maximum
    angle = 10 # Initial angle for the spiral rotation
    size = 100 # Initial size for the heptagons

    def draw_heptagon(size):
        """
        Draws a heptagon with the given size.
        """
        for _ in range(7):
            t.forward(size) # Draw each side of the heptagon
            t.left(360 / 7) # Turn to the next side

    # Draw the spiral pattern of heptagons
    while size > 10:
        # Set the fill color with a gradient from green to red
        t.fillcolor((1 - size / 100, size / 100, 0))
        t.begin_fill() # Start filling the shape
        draw_heptagon(size) # Draw the heptagon
        t.end_fill() # End filling the shape
        t.right(angle) # Rotate right to create the spiral
        effect
        size -= 5 # Decrease the size of the heptagon for the
        next iteration
        t.left(10) # Adjust the rotation slightly to maintain
        the spiral pattern
```

(c) Generated CoT label for the first image ⬢ in Figure 10b.

Figure 10: Examples of images in the seed dataset and the TURTLEAI-Train dataset. The seed dataset include 10 task images and their corresponding solution codes. The TURTLEAI-Train include 738k images generated from the seed dataset using our data generation framework TURTLEAI-Datagen. TURTLEAI-Datagen can generate diverse and high-quality images by evolving from a small seed dataset.

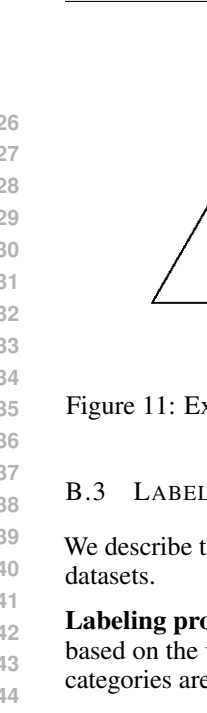 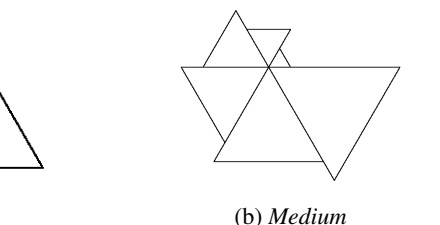 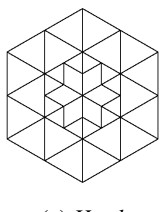

(a) *Easy*  (b) *Medium*  (c) *Hard*

Figure 11: Examples showing images of different difficulty levels from the TURTLEAI-DS dataset.

### B.3 LABELING PROCESS

We describe the dataset labeling process for the task categories and difficulty levels in the evaluation datasets.

**Labeling process of task categories.** We identify 6 different geometric categories of task images based on the visual patterns and transformations in the evaluation datasets. The definitions of these categories are as follows:

- *Basic geometry*: simple and basic shapes like squares, circles, triangles, and lines without complex patterns or arrangements. Tasks in this category require understanding of basic geometry.

- *Rotation*: patterns formed by rotating basic geometric shapes around a central point to create symmetrical designs, such as a spirograph. These tasks require reasoning about rotation angles and the numbers of repetitions.

- *Translation*: patterns formed by translating a basic pattern to different positions, forming tiling or grid structures. Tasks in this category require reasoning about translation distance and the numbers of basic patterns.

- *Scaling*: patterns formed by scaling basic geometric shapes, creating nested or expanding structures. Tasks in this category require reasoning about scaling factors and the numbers of repetitions.

- *Spiral*: sequential shapes arranged in spiraling paths, creating dynamic patterns with radial symmetry, like an Archimedean spiral. Tasks in this category require reasoning about the spiral pattern, the numbers of repetitions, the degrees of rotation, and scaling factors.

- *Composite*: complex arrangements combining different transformations (scaling, rotation, spiral, and translation) with varied shapes or colors.

Given above definitions, we manually label each task-code pair in the dataset TURTLEAI-DS into one of the 6 categories. Note that an image may involve multiple transformations, we categorize each image based on its predominant geometric characteristic.

**Labeling process of difficulty levels.** We assigned a difficulty level for each task-code pair in the dataset TURTLEAI-DS. The assigned difficulty level is based on the complexity of the visual patterns, and difficulty of drawing the image using Turtle Graphics. We define 3 difficulty levels as follows:

- *Easy*: basic patterns consisting of single geometric shapes like squares, circles, or lines without complex combinations. Tasks in this category require entry-level visual understanding of geometry, basic math reasoning about transformation parameters (e.g., length, angles), and fundamental programming skills to implement simple geometric shapes.

- *Medium*: patterns involving combinations of basic shapes with one or more transformations. Tasks in this category require the ability to decompose patterns into simpler components, understand relationships between different transformations, and programming skills to implement multiple transformation steps.

- *Hard*: sophisticated patterns featuring multiple complex transformations. Tasks in this category require high-level visual understanding, reasoning about spatial and temporal relationships between components, mathematical reasoning about transformation parameters

(e.g., length, angles, scaling factors), and programming skills to convert complex reasoning into executable programs.

Given above definitions, we manually label each image in the dataset TURTLEAI-DS into one of the 3 difficulty levels.

**Labeling process of failure types.** We identify 7 different failure types based on the failure cases of different models on the TURTLEAI-DSBasic dataset. The definitions of these failure types are as follows:

- *Visual understanding*: This involves misinterpreting the image's overall design, layout, or composition, leading to code generation that does not match the target image. For instance, the image shows a square, but the model describes it as a circle and generates the code for a circle.

- *Decomposition*: This involves errors in decomposing the image into feasible drawing steps or errors in decomposing a complex pattern into its constituent parts (shapes, patterns, elements). For example, when a complex pattern results from repeated rotations of a simple shape. The model may fail to decompose it into the base shape and its transformations, instead treating the entire pattern as an indivisible unit. This leads to either an attempt to generate code that represents the entire complex pattern directly or a failure to plan a feasible sequence of steps to generate the pattern.

- *Spatial reasoning*: This involves errors in understanding relative positions, distances, angles, and sizes of patterns within the image. For instance, the model misinterprets the relative positioning of two shapes, placing one above the other when they are actually side-by-side.

- *Programming*: This involves the model having correct visual understanding and reasoning but the code implementation is not consistent with the visual reasoning results or the code implementation contains syntax or logical errors. For instance, the model understands correctly that the image show a square, but implements the code for a circle instead.

- *Visual precision*: This involves the model having correct visual understanding and reasoning, but failing to achieve very precise details during the code implementation. For instance, the model generates code that captures the overall structure but deviates in specifics, such as lines being slightly too long, angles that are a few degrees off.

- *Repetition*: This involves unnecessary or incorrect repetition of code blocks. For instance, the model keeps generating the same redundant code repeatedly without stopping.

- *Evaluation error*: This is due to the evaluation framework's incorrect evaluation results that are not consistent with the manual evaluation results. For instance, the symbolic comparison incorrectly identifies the generated image as a *success*, but it's actually a *fail* from the manual evaluation.

# C  ADDITIONAL EXPERIMENTS AND ANALYSIS

In this section, we provide additional experiments and analysis of our synthetic data generation framework TURTLEAI-Datagen and the model performance on TURTLEAI.

## C.1  ANALYSIS OF THE SIZE OF THE DATASET GENERATED BY TURTLEAI-DATAGEN

We analyze the exponential growth rate of datasets generated by the TURTLEAI-Datagen framework. Specifically, we derive a mathematical formulation that describes how the dataset size expands iteratively, starting from an initial seed dataset and growing with each iteration. Formally, assume that we have an initial seed dataset $\mathcal{D}_0$ containing $|\mathcal{D}_0|$ samples. Starting from $\mathcal{D}_0$, the framework generates a dataset $\mathcal{D}_t$ after $t$ iterations, where the size $|\mathcal{D}_t|$ can be expressed as:

$$|\mathcal{D}_t| = |\mathcal{D}_{t-1}| \cdot p \cdot k \cdot (1 - d_{t-1}) \tag{3}$$

$$= |\mathcal{D}_0| \cdot (pk)^t \cdot \prod_{i=1}^{t}(1 - d_{i-1}), \tag{4}$$

where $p \geq 1$ is the number of pairs of codes sampled from the dataset in each iteration as the reference-guided code-to-code mutation example (e.g., $p = 4, 8, 16$), $d_i \in [0, 1]$ is the duplicate rate in the $i$-th iteration, and $k \in [0, 1]$ is the top percentage of samples selected from the dataset in the elite selection stage. If we assume that the duplicate rate is the same for all iterations (i.e., $d_i = d$), then the size of the dataset $|\mathcal{D}_t|$ can be simplified as:

$$|\mathcal{D}_t| = |\mathcal{D}_0| \cdot \big(pk(1 - d)\big)^t, \tag{5}$$

where the size of the dataset $|\mathcal{D}_t|$ grows exponentially with the number of iterations $t$ with a growth rate of $pk(1-d)$. In our implementation, we set $p = 16$ and $k = 0.7$ for generating TURTLEAI-Train, resulting in a growth rate of around 9.7 at each iteration.

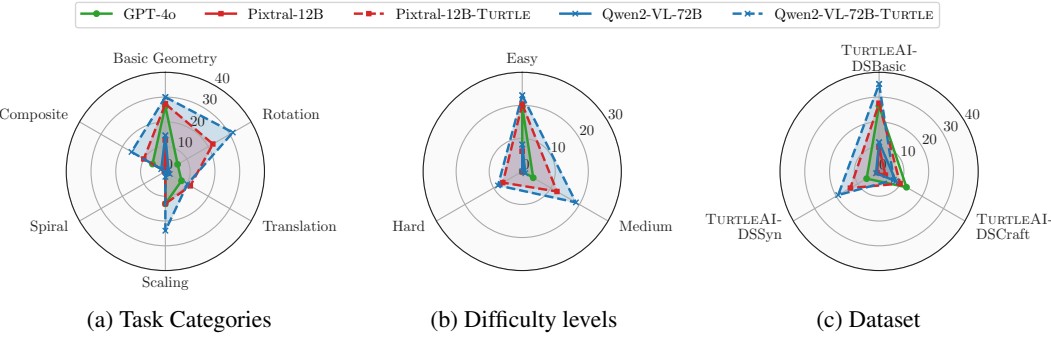

(a) Task Categories   (b) Difficulty levels   (c) Dataset

Figure 12: Performance comparison of VLMs across different task categories, difficulty levels, and datasets. We use symbolic-based success rate as the metric. Evaluated VLMs struggle most in categories requiring spatial transformations and their performance decreases as the difficulty level increases.

## C.2  ANALYSIS OF MODEL PERFORMANCE ACROSS DIFFERENT DIMENSIONS

To identify strengths and weaknesses, we compare the performance of several representative VLMs across different task categories, difficulty levels, and datasets, providing a more nuanced analysis. The performance comparison is shown in Figure 12.

In Figure 12a, we compare the performance of different models across different task categories. Our results show that existing models perform relatively better on *Basic geometry* tasks but struggle with categories that require spatial transformations, such as *Translation*, *Scaling*, and *Rotation*. Performance deteriorates further for *Spiral* and *Composite* tasks, which necessitate combinations of multiple transformations and complex color usage, making them particularly challenging for existing models. In Figure 12b, we compare the performance of different models across different

| | TURTLEAI-DSBasic | | TURTLEAI-DS | |
|---|---|---|---|---|
| | CoT | Non-CoT | CoT | Non-CoT |
| Pixtral-Large | **11.76** | 10.78 | 4.74 | **6.56** |
| Qwen2-VL-72B | 10.78 | **11.76** | **4.13** | 3.52 |
| Llava-OneVision-72B | **8.82** | 4.90 | **2.19** | **2.19** |
| InternVL2-76B | 8.82 | **11.76** | 2.79 | **3.28** |
| Molmo-72B | **9.80** | 3.92 | **2.92** | 2.31 |
| Pixtral-12B | 1.96 | **9.80** | 1.09 | **2.31** |
| Llava-OneVision-7B | **3.92** | **3.92** | 0.97 | **1.09** |
| Qwen2-VL-7B | **4.90** | 0.98 | **1.09** | 0.12 |
| Molmo-7B | 0.00 | 0.00 | **0.12** | 0.00 |
| InternVL2-8B | **2.94** | 0.00 | **0.73** | 0.12 |

Figure 13: Symbolic success rates of different base VLMs on TURTLEAI-DSBasic and TURTLEAI-DS with and without CoT prompting. The best performance is highlighted in **bold** for each model.

difficulty levels. Our results show that model performance consistently decreases as the difficulty level increases. Figure 12c shows the performance of different models across different datasets. We find that models generally perform better on TURTLEAI-DSBasic than TURTLEAI-DSSyn due to the higher difficulty of the TURTLEAI-DSSyn dataset. When comparing the TURTLEAI-DSBasic and the TURTLEAI-DSCraft, we find that models generally perform better on the TURTLEAI-DSBasic dataset due to the introduced variations in the hand-drawn dataset TURTLEAI-DSCraft. Interestingly, our fine-tuned models are also comparable or even perform better than the corresponding base models on TURTLEAI-DSCraft. For instance, Pixtral-12B-TURTLE performs better than Pixtral-12B on TURTLEAI-DSCraft.

### C.3 INFLUENCE OF THE CoT PROMPTING ON MODEL PERFORMANCE

We investigate the influence of the CoT prompting on base models' performance. We experiment with various open-source VLMs, with and without CoT prompting. For CoT prompting, we require the model to generate the solution code in the following step-by-step manner: (i) describe the image in detail, (ii) analyze the image and propose steps to create the pattern, and (iii) generate the solution code with comprehensive docstrings and comments. For non-CoT prompting, we only require the model to generate the code, without the above steps explicitly mentioned. The comparison results are shown in Figure 13. The results indicate that the effectiveness of CoT prompting varies across different models and datasets, and there is no clear indication that CoT prompting can improve performance in our domain. For instance, Qwen2-VL-7B shows improved performance with CoT prompting on both datasets, whereas InternVL2-76B performs better without CoT prompting on both datasets. This inconsistency may stem from the reasoning-intensive nature of our tasks, where each type of task demands different reasoning steps, making it challenging to devise a consistent CoT prompting strategy applicable to all tasks. Furthermore, models trained on different datasets may develop distinct reasoning preferences, causing the same CoT strategy to enhance performance in some models while potentially confusing others, resulting in inconsistent performance of CoT prompting in our domain.

### C.4 INFLUENCE OF LoRA RANK AND VISION TOWER FOR FINE-TUNING PERFORMANCE

We investigate the influence of LoRA rank and vision tower fine-tuning on model performance. To do this, we conduct fine-tuning experiments on Pixtral-12B with LoRA ranks of 64, 128, and 256 using the 738k TURTLEAI-Train dataset (without CoT labeling), training each configuration for 1 epoch. For each LoRA rank, we set the LoRA alpha parameter to twice the rank value. Additionally, we examine the impact of freezing versus unfreezing the vision tower during fine-tuning. By unfreezing the vision tower, we enable parameter tuning of the visual encoder component of the VLM, allowing the model to adapt its visual representations during fine-tuning. The results are shown in Figure 14. We find that unfreezing the vision tower can enhance performance. Specifically, in our experiments with LoRA rank 64, unfreezing the vision tower increases the success rate from $10.78\%$ to $17.65\%$

| | Fine-tuning Parameters | | Success Rate (%) | |
|---|---|---|---|---|
| | Vision Tower | LoRA rank | TURTLEAI-DSBasic | TURTLEAI-DS |
| Pixtral-12B-TURTLE | Freeze | 64 | 10.78 | 8.38 |
| Pixtral-12B-TURTLE | Unfreeze | 64 | 17.65 | 9.96 |
| Pixtral-12B-TURTLE | Unfreeze | 128 | **22.55** | **12.67** |
| Pixtral-12B-TURTLE | Unfreeze | 256 | 15.69 | 10.81 |

Figure 14: Influence of LoRA rank and vision tower on the performance of fine-tuning. We experiment with LoRA ranks of 64, 128, and 256, freezing and unfreezing the vision tower to fine-tune Pixtral-12B model using the 738k TURTLEAI-Train dataset (without CoT labeling), with each setting trained for 1 epoch. Unfreezing the vision tower and using LoRA rank 128 yields the best performance.

on TURTLEAI-DSBasic and from $8.38\%$ to $9.96\%$ on TURTLEAI-DS. Additionally, the choice of LoRA rank also affects the performance, with rank 128 achieving the best results in our case.

## C.5 PERFORMANCE OF VLMS USING PASS@K METRICS

| | | Greedy Decoding | Random Sampling | | |
|---|---|---|---|---|---|
| | Size | Pass@1 | Pass@1 | Pass@3 | Pass@5 |
| InternVL2 | 76B | 3.28 | 2.72 | 4.79 | 5.83 |
| Llava-OneVision | 72B | 2.19 | 2.09 | 4.02 | 5.35 |
| Qwen2-VL | 72B | 3.52 | 3.18 | 5.53 | 6.93 |
| Qwen2-VL-TURTLE | 72B | 19.56 | 16.79 | 25.65 | 29.77 |
| InternVL2 | 8B | 0.12 | 0.19 | 0.51 | 0.73 |
| Llava-OneVision | 7B | 1.09 | 0.83 | 1.65 | 2.19 |
| Qwen2-VL | 7B | 0.12 | 0.10 | 0.29 | 0.49 |
| Qwen2-VL-TURTLE | 7B | 13.37 | 11.96 | 19.77 | 23.69 |
| Pixtral | 12B | 2.31 | 1.48 | 3.23 | 4.25 |
| Pixtral-TURTLE | 12B | 14.70 | 10.28 | 17.36 | 20.66 |

Figure 15: Symbolic success rates (%) of VLMs on the dataset TURTLEAI-DS with greedy decoding and random sampling. For greedy decoding, we report Pass@1 using `temperature = 0`. For random sampling, we use `temperature = 0.8` and `top_p = 0.95`, where for each Pass@K metric, we generate $N = 5$ samples.

In the main paper, we report the evaluation results of different VLMs on our benchmark using a greedy decoding strategy (i.e., `temperature=0`). To provide a more comprehensive evaluation, we also experiment with a random sampling strategy by randomly sampling $N$ samples from the model and then calculating the Pass@K results. Following previous works (Rozière et al., 2023; Zhuo et al., 2025), we compute Pass@K results with random sampling by generating $N = 5$ samples with `top_p=0.95` and `temperature = 0.8`. Then we calculate Pass@1, Pass@3, and Pass@5, respectively. Although generating many more samples ($N \geq K$) is recommended to reduce bias, we adopt the lower bound due to limited computational resources. The results are provided in Figure 15.

# D  RELIABILITY OF THE EVALUATION FRAMEWORK

|  | Positive (Symbolic) | Negative (Symbolic) |
|---|---|---|
| Positive (Manual) | 74 | 5 |
| Negative (Manual) | 2 | 742 |

$Precision = 0.974$    $Recall = 0.937$    $F1 = 0.955$    $Accuracy = 0.991$

(a) Confusion matrix for the symbolic comparison.

|  | Positive (Embedding) | Negative (Embedding) |
|---|---|---|
| Positive (Manual) | 69 | 10 |
| Negative (Manual) | 6 | 738 |

$Precision = 0.873$    $Recall = 0.920$    $F1 = 0.896$    $Accuracy = 0.981$

(b) Confusion matrix for the embedding-based comparison.

Figure 16: Confusion matrices illustrating the accuracy of our evaluation framework by comparing the results of the symbolic and embedding-based comparisons against the manual comparison. The evaluation is conducted by manually annotating GPT-4o's results on the TURTLEAI-DS dataset. (a) shows the confusion matrix for the symbolic comparison, which demonstrates high accuracy with an F1 score of 0.955 when compared against manual evaluation. (b) shows the confusion matrix for the embedding-based comparison, achieving an F1 score of 0.896 at a threshold value of 0.95.

To assess the reliability of our evaluation framework, we perform manual evaluation and compare it with the accuracies of both symbolic and embedding-based comparisons. To do this, we first perform a manual evaluation of all generated images in the TURTLEAI-DS dataset from GPT-4o. This involves comparing the ground-truth image with the corresponding image produced by executing the generated code from GPT-4o, and manually verifying whether the each generated image is visually identical to the ground-truth image. This manual evaluation involves a total of 823 image-code pairs. After this manual evaluation, we compare our results with both symbolic and embedding-based comparisons to evaluate the accuracy of our evaluation framework.

**Accuracy of the symbolic comparison.** After manual evaluation, we use our manual evaluation results as ground truth and calculate the precision, recall, F1 score, and accuracy for the results of the symbolic comparison. The results are shown in Figure 16a. Our symbolic comparison achieves a precision of 0.974, recall of 0.937, F1 score of 0.955, and accuracy of 0.991, showing that the symbolic comparison can correctly identify almost all of the generated images compared against the manual evaluation.

**Accuracy of the embedding-based comparison.** The embedding-based comparison first calculates a similarity score and then determines *success* or *fail* by comparing the similarity score against a threshold value. To determine the optimal threshold value, we plot how different threshold values affect the precision, recall, and F1 score, and then we select the threshold value that maximizes the F1 score. Figure 17 shows the precision, recall, and F1 score at different threshold values. We find that using a threshold of 0.95 achieves the highest F1 score of 0.896. Therefore, we use a threshold of 0.95 for the embedding-based comparison, i.e., if the embedding score is greater than 0.95, we consider the generated image as a *success*. Figure 16b shows detailed statistics of the precision, recall, F1 score, and accuracy for the embedding-based comparison with a threshold of 0.95.

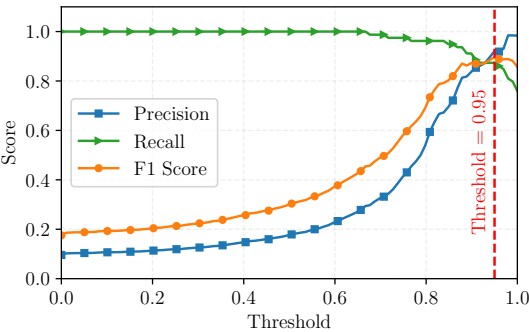

Figure 17: The relationship between precision, recall, and F1 score at different thresholds used in the embedding-based comparison. The best F1 score is achieved at a threshold of 0.95, with F1 score of 0.896.

# E IMPLEMENTATION DETAILS

In this section, we detail the implementation of our dataset generation framework TURTLEAI-Datagen, model fine-tuning process, evaluation process, and the evaluation framework TURTLEAI-Eval.

## E.1 IMPLEMENTATION DETAILS OF TURTLEAI-DATAGEN

We describe the implementation details of the dataset generation framework TURTLEAI-Datagen.

**Stage 1: code mutation.** We use the Llama3.1-70B-Instruct model for code mutation. The model is queried with `temperature = 0.5` and `top_p = 1`. We use higher temperature and `top_p` values to encourage the model to generate more diverse and creative code. During code mutation, we randomly sample 16 pairs of $(C_{ref_1}, C_{ref_2})$ from the seed dataset for each input code $C_{in}$. This results in 16 possible mutated codes for $C_{in}$ after applying the mutation for each pair of $(C_{ref_1}, C_{ref_2})$. An illustrative example of the code mutation process is provided in Figure 18.

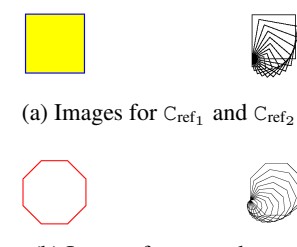

(a) Images for $C_{ref_1}$ and $C_{ref_2}$

(b) Images for $C_{in}$ and $C_{out}$

```
- Introduce a loop: Add a loop to create
multiple instances of a shape with changing
parameters.
- Parameterize shape attributes:
Parameterize attributes such as size,
angle, or color to create a dynamic and
changing pattern.
- Focus on the pattern: Remove or modify
existing features to focus on the new
pattern created by the loop.
```

(c) Inferred mutation pattern $m(C_{ref_1}, C_{ref_2})$

Figure 18: An illustrative example for the reference-guided code mutation. (a) shows the corresponding images for a pair of sampled reference codes $(C_{ref_1}, C_{ref_2})$. (b) shows the corresponding images for $C_{in}$ and the corresponding mutated code $C_{out}$ by the LLM's inferred mutation pattern in (c).

**Stage 2: elite selection.** The elite selection stage consists of two steps: deduplication and selection of elite samples. Given a dataset consisting of image-code pairs, we first perform deduplication to remove duplicate images. Specifically, we use a pre-trained ResNet18 (He et al., 2016) image encoder to obtain the embedding for each image. Then we use the DBSCAN clustering algorithm (Ester et al., 1996) to cluster these image embeddings, resulting in a set of clusters, where each cluster consists of similar images. For each cluster, we only preserve one sample and remove the rest. We use DBSCAN clustering algorithm with parameters $\epsilon = 0.2$, `min_samples = 2`, and the `euclidean` distance. After deduplication, we select the elite samples from the deduplicated dataset. To do this, we use Qwen2-VL-72B as the model for selecting elite samples. We use top $k = 30\%$ for generating the TURTLEAI-DSSyn dataset and $k = 70\%$ for generating the TURTLEAI-Train dataset.

**Stage 3: CoT labeling.** We use Pixtral-Large as the model for CoT labeling stage. The model is queried with `temperature = 0.1` and `top_p = 0.001`. Note that this stage is only used to generate the TURTLEAI-Train dataset for fine-tuning.

For querying above models in different stages, we consistently use the vLLM inference engine to speed up the inference. During the inference, we use $8 \times$ H100 GPUs, with `tensor_parallel_size` set to 8, and the `max_num_seqs` set to 64. For every 100k samples generated during the elite selection (using Qwen2VL-72B-Instruct) or code mutation stage (using Llama3.1-70B-Instruct), the process takes approximately 8 hours. For the CoT labeling (using Pixtral-Large), it takes around 13 hours to process every 100k samples.

## E.2 IMPLEMENTATION DETAILS OF FINE-TUNING

We conduct fine-tuning experiments on three models: Qwen2-VL-7B (Wang et al., 2024), Qwen2-VL-72B (Wang et al., 2024), and Pixtral-12B (Agrawal et al., 2024). In our fine-tuning experiments, we use LoRA (Hu et al., 2022) for parameter-efficient fine-tuning. To determine the best LoRA rank and scaling factor, we experimented with LoRA ranks of 64, 128, and 256, using a scaling factor two times the LoRA rank in each case. We found that a rank of 128 provides the best performance. Consequently, we use a LoRA rank of 128 and a scaling factor of 256 for all fine-tuning experiments. We also experimented with freezing and unfreezing the vision tower during fine-tuning and found that

| Model | Version |
|---|---|
| o3 | o3-2025-04-16 (`reasoning_effort=medium`) (OpenAI, 2025b) |
| o4-mini | o4-mini-2025-04-16 (`reasoning_effort=medium`) (OpenAI, 2025b) |
| GPT-5 (medium) | gpt-5-2025-08-07 (`reasoning_effort=medium`) (OpenAI, 2025a) |
| GPT-4o | gpt-4o-2024-11-20 (OpenAI, 2024a) |
| GPT-4V | gpt-4-turbo-2024-04-09 (OpenAI, 2024b) |
| Qwen2-VL-72B | Qwen/Qwen2-VL-72B-Instruct (Wang et al., 2024) |
| Qwen2-VL-7B | Qwen/Qwen2-VL-7B-Instruct (Wang et al., 2024) |
| Qwen2-VL-72B-TURTLE | Qwen2-VL-72B-Instruct (fine-tuned on TURTLEAI-Train) |
| Qwen2-VL-7B-TURTLE | Qwen2-VL-7B-Instruct (fine-tuned on TURTLEAI-Train) |
| Molmo-72B | allenai/Molmo-72B-0924 (Deitke et al., 2024) |
| Molmo-7B | allenai/Molmo-7B-D-0924 (Deitke et al., 2024) |
| Llava-OneVision-72B | llava-hf/llava-onevision-qwen2-72b-ov-chat-hf (Li et al., 2024a) |
| Llava-OneVision-7B | llava-hf/llava-onevision-qwen2-7b-ov-chat-hf (Li et al., 2024a) |
| NVLM-1.0-D | nvidia/NVLM-D-72B (Dai et al., 2024) |
| Pixtral-Large | mistralai/Pixtral-Large-Instruct-2411 (Agrawal et al., 2024) |
| Pixtral-12B | mistral-community/pixtral-12b (Agrawal et al., 2024) |
| Pixtral-12B-TURTLE | Pixtral-12B (fine-tuned on TURTLEAI-Train) |
| InternVL2-76B | OpenGVLab/InternVL2-Llama3-76B (Chen et al., 2023) |
| InternVL2-8B | OpenGVLab/InternVL2-8B (Chen et al., 2023) |
| GLM-4V-9B | THUDM/glm-4v-9b (Zeng et al., 2024b) |

Figure 19: Evaluated models and their versions. Highlighted models are our fine-tuned models using TURTLEAI-Train.

unfreezing the vision tower provides better performance. Therefore, we unfreeze the vision tower during fine-tuning for all fine-tuning experiments.

During fine-tuning, we use a learning rate schedule that combines a $10\%$ warmup phase where the learning rate linearly increases to $1e - 4$, followed by cosine annealing which gradually reduces the learning rate to $0$, ensuring stable training and smooth convergence (Zheng et al., 2024). We also reserve $1\%$ of the 738k training dataset for validation. For our fine-tuning experiments on Qwen2-VL-7B and Pixtral-12B models, we observed that validation loss increases after the first epoch, so we stopped the fine-tuning after one epoch and report their performance at epoch 1 accordingly. Conversely, the Qwen2-VL-72B model's performance continued to improve after the first epoch, so we fine-tuned it for two epochs in total.

All fine-tuning experiments are conducted on an internal cluster using $8 \times$ H100 GPUs, with each epoch taking approximately $15$ hours for Qwen2-VL-7B, $112$ hours for Qwen2-VL-72B, and $22$ hours for Pixtral-12B.

### E.3 IMPLEMENTATION DETAILS OF EVALUATION

**Inference details of VLMs.** For open-source VLMs, we download their pre-trained weights and perform inference locally. We use the vLLM (Kwon et al., 2023) engine for VLM inference to obtain the outputs of the evaluated open-source VLMs. During inference, we set the `temperature` to 0 and use different numbers of GPUs for different models depending on their parameter sizes: (i) $1 \times$ A100 GPU for models with parameter sizes less than 7B, (ii) $2 \times$ A100 GPUs for models with parameter sizes between 7B and 70B, and (iii) $8 \times$ A100 GPUs for models with parameter sizes larger than 70B. We set the vLLM parameter `tensor_parallel_size` to 1, 2, and 8 for the three cases, respectively, and set `max_num_seqs` to `tensor_parallel_size` $\times$ 8. We use the OpenAI API to evaluate proprietary models from OpenAI. For reasoning models, we set `reasoning_effort` to `medium` and `max_completion_tokens` to 8192.

**Details of the evaluation procedure.** For each task in our evaluation datasets, we provide the task image along with a fixed prompt template (see Figure 29) to guide the VLMs in generating Turtle Graphics Python code. The model's output often includes an explanation along with the predicted

code. We extract only the code snippet and disregard the rest. If multiple code snippets are present, we handle them differently based on the comparison method:

- *Symbolic comparison*: we evaluate all the code snippets and consider the result a *success* if any code snippet is successful.

- *Embedding-based comparison*: we evaluate only the longest code snippet. This is because embedding-based comparison involves batch processing when extracting image embeddings, making it inefficient to consider multiple code snippets.

### E.4    IMPLEMENTATION DETAILS OF THE EVALUATION FRAMEWORK

We describe the implementation details of our evaluation framework. Given a task image `img`, the predicted code snippet $\hat{C}$, and the solution code `C`, our evaluation framework works as follows to evaluate the correctness of the predicted code $\hat{C}$. First, we execute both the solution code `C` and the predicted code $\hat{C}$ using a customized Turtle Graphics emulator. This emulator inherits the built-in Turtle Graphics module and enables us to record all the drawing states, including the coordinates and the colors when drawing a line, filling a polygon, etc. Second, we transform the recorded drawing states for `C` and $\hat{C}$ into the same space to ensure the invariance to size, position, and line width of the drawing. Specifically, we perform the following three steps:

- *Normalizing length of lines*: We rescale all recorded coordinates such that the maximum dimension of the entire pattern's bounding box is set to 300. This ensures that the drawings are uniformly scaled regardless of their original size, making our comparison invariant to the size of the drawing.

- *Centering around the origin*: We translate all recorded lines so that they are centered around the origin. This involves calculating the center of the bounding box and shifting all coordinates accordingly. This ensures that the comparison is invariant to the position of the drawing.

- *Standardizing pen size*: We standardize the pen size of all lines to 1. This ensures that drawing line width does not affect the comparison of drawings, making our comparison invariant to the line width.

Third, we render these normalized drawing states into images in the sequence as they are recorded, resulting in standardized images `img` and $\hat{\text{img}}$ for `C` and $\hat{C}$, respectively. Finally, our evaluation framework provides two comparison methods to compare these two images, which are described in detail as follows.

**Symbolic comparison.** This compares the standardized images `img` and $\hat{\text{img}}$ pixel-by-pixel. The high-level idea is to first count non-white pixels in both images and calculate the percentage of differing pixels among them. If this percentage is below a predefined value, the comparison result is *success*; otherwise, the comparison result is *fail*. More specifically, assume the images `img` and $\hat{\text{img}}$ are of dimensions $H \times W$ and $\text{img}_{i,j}$ and $\hat{\text{img}}_{i,j}$ are the pixels at position $(i, j)$, respectively. We define a candidate set of pixels $\mathcal{P}$ that are considered for symbolic comparison:

$$\mathcal{P} = \big\{ (i,j) \mid \text{img}_{i,j} \neq \text{white} \vee \hat{\text{img}}_{i,j} \neq \text{white}, \tag{6}$$

$$\forall\, i \in \{1, \ldots, H\},\, j \in \{1, \ldots, W\} \big\}. \tag{7}$$

The pixel-wise difference between `img` and $\hat{\text{img}}$ is computed as:

$$\text{pixel\_diff}(\text{img}, \hat{\text{img}}) = \frac{\sum_{(i,j) \in \mathcal{P}} \mathbb{I}(\text{img}_{i,j} \neq \hat{\text{img}}_{i,j})}{|\mathcal{P}|}, \tag{8}$$

where $\mathbb{I}(\text{img}_{i,j} \neq \hat{\text{img}}_{i,j})$ is the indicator function that returns 1 if $\text{img}_{i,j} \neq \hat{\text{img}}_{i,j}$ and 0 otherwise. We establish a threshold for pixel-wise differences to determine whether the image $\hat{\text{img}}$ is a *success* to match `img` or not. If $\text{pixel\_diff}(\text{img}, \hat{\text{img}}) < 1 - \text{threshold}$, the image $\hat{\text{img}}$ is considered a *success* to match `img`; otherwise, it is considered a *fail*. For our symbolic evaluation, we use a threshold $= 0.95$ for drawings with fill colors and $0.92$ for those without fill colors. The lower threshold for filled drawings (i.e., using `begin_fill()` and `end_fill()` in the code) is due to

the typically larger candidate set $\mathcal{P}$ for these drawings. The pixel-wise similarity in filled areas can overshadow differences in non-filled areas, making them harder to detect. Thus, we set a stricter threshold for filled drawings.

**Embedding-based comparison.** This method compares the standardized images $\texttt{img}$ and $\hat{\texttt{img}}$ within the embedding space. This is achieved by extracting image embeddings from both $\texttt{img}$ and $\hat{\texttt{img}}$ using a pre-trained image encoder model and then calculating a similarity score between these embeddings using a distance metric. During implementation, when comparing two standardized images $\texttt{img}$ and $\hat{\texttt{img}}$, we first resize them to 256x256 pixels, apply a center crop to 224x224 pixels, convert them to tensors, and normalize them using standard ImageNet statistics to ensure consistency and accuracy. Then we extract 512-dimensional feature vectors from these images using the ResNet18 model pre-trained on ImageNet (He et al., 2016).[8] The similarity score between these embeddings is computed using the Euclidean distance, normalized to the range $[0, 1]$, where a higher score indicates higher similarity in the embedding space. Images are processed in batches of 128 for efficient computation. Any pairs where image processing fails (e.g., empty images or images that are too large) are assigned a similarity score of 0. Finally, we search the optimal threshold $= 0.95$ because it achieves the best F1 score (see Figure 17). If the similarity score exceeds 0.95, the image $\hat{\texttt{img}}$ is considered a *success* in matching $\texttt{img}$; otherwise, it is considered a *fail*.

---

[8]We use ResNet-18 primarily because it is a widely adopted, well-performing, and lightweight model (with only 11.7 million parameters) that offers a reasonable trade-off between performance and speed.

| | GPT-4o | Pixtral-12B Qwen2-VL-72B | Pixtral-12B-TURTLE Qwen2-VL-72B-TURTLE | # Cases | Percentage |
|---|---|---|---|---|---|
| Figure 21 | ✓ | ✓ | ✓ | 6 | 0.73% |
| N.A. | ✗ | ✓ | ✓ | 0 | 0.00% |
| Figure 22 | ✓ | ✗ | ✓ | 17 | 2.06% |
| Figure 23 | ✓ | ✓ | ✗ | 3 | 0.36% |
| Figure 24 | ✓ | ✗ | ✗ | 23 | 2.79% |
| N.A. | ✗ | ✓ | ✗ | 0 | 0.00% |
| Figure 25 | ✗ | ✗ | ✓ | 54 | 6.56% |
| Figure 26 | ✗ | ✗ | ✗ | 591 | 71.81% |

Figure 20: Summary of different possible failure cases across different types of models on our benchmark tasks. We identify eight possible failure cases for the GPT-4o, open-source base models (Pixtral-12B and Qwen2-VL-72B), and fine-tuned models (Pixtral-12B-TURTLE and Qwen2-VL-72B-TURTLE). For each possible failure case, we provide the number of occurrences and the corresponding percentage, with references to detailed examples in corresponding figures. Success and failure are indicated by ✓ and ✗, respectively.

# F    CASE STUDY OF FAILURES

We provide a case study of different models' outputs on tasks in the TURTLEAI-DS dataset. We select five representative VLMs: GPT-4o, Pixtral-12B, Qwen2-VL-72B, Pixtral-12B-TURTLE, and Qwen2-VL-72B-TURTLE. To systematically analyze the failure cases, we enumerate all possible failure cases across different types of models and provide examples for each type.

Specifically, we categorize these 5 models into 3 types: proprietary model (GPT-4o), open-source base models (i.e., Pixtral-12B, Qwen2-VL-72B), and fine-tuned models (Pixtral-12B-TURTLE, Qwen2-VL-72B-TURTLE). Then we categorize the failure cases into 8 different possibilities and show examples for each possibility. These possibilities are summarized in Figure 20. Figure 27 and Figure 28 show example model responses.

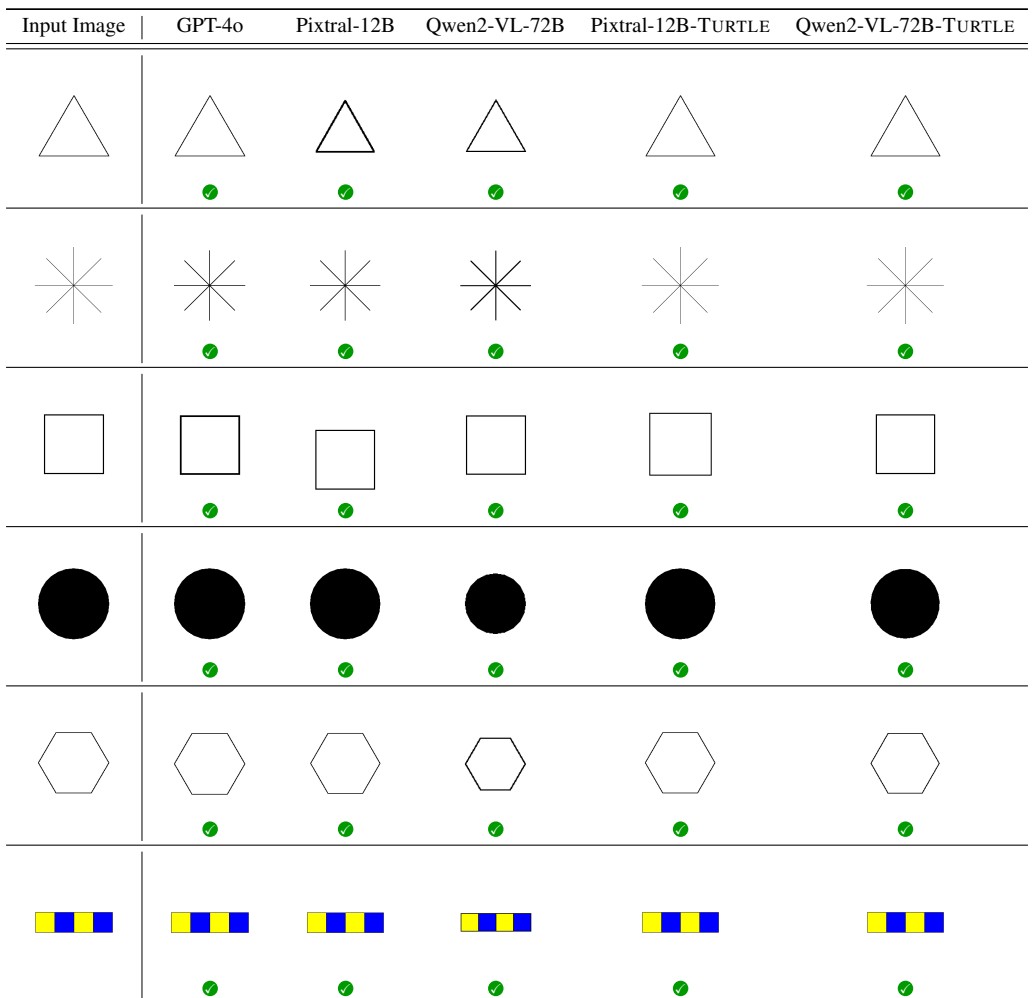

Figure 21: Tasks that are successfully solved by base models (i.e., GPT-4o, Pixtral-12B, Qwen2-VL-72B) and our fine-tuned models (i.e., Pixtral-12B-TURTLE, Qwen2-VL-72B-TURTLE) in the TURTLEAI-DS dataset with 823 tasks. A total of 6 tasks (0.73%) match this criteria. Each row shows a ground truth image (leftmost) followed by the corresponding images generated by executing the each model's generated Python code. Success (✓) and failure (✗) are determined by our evaluation framework using symbolic comparison.

## G  PROMPTS

In this section, we provide the prompts used in the TURTLEAI as follows:

- Figure 29 shows the prompt used for guiding VLMs in synthesizing code from a given image input.
- Figure 30 shows the prompt for the reference-guided code generation stage in our data synthesis framework TURTLEAI-Datagen.
- Figure 31 provides the prompt used for the elite selection stage in TURTLEAI-Datagen for scoring the quality of the generated geometric image.
- Figure 32 provides the prompt for the CoT labeling stage in TURTLEAI-Datagen.

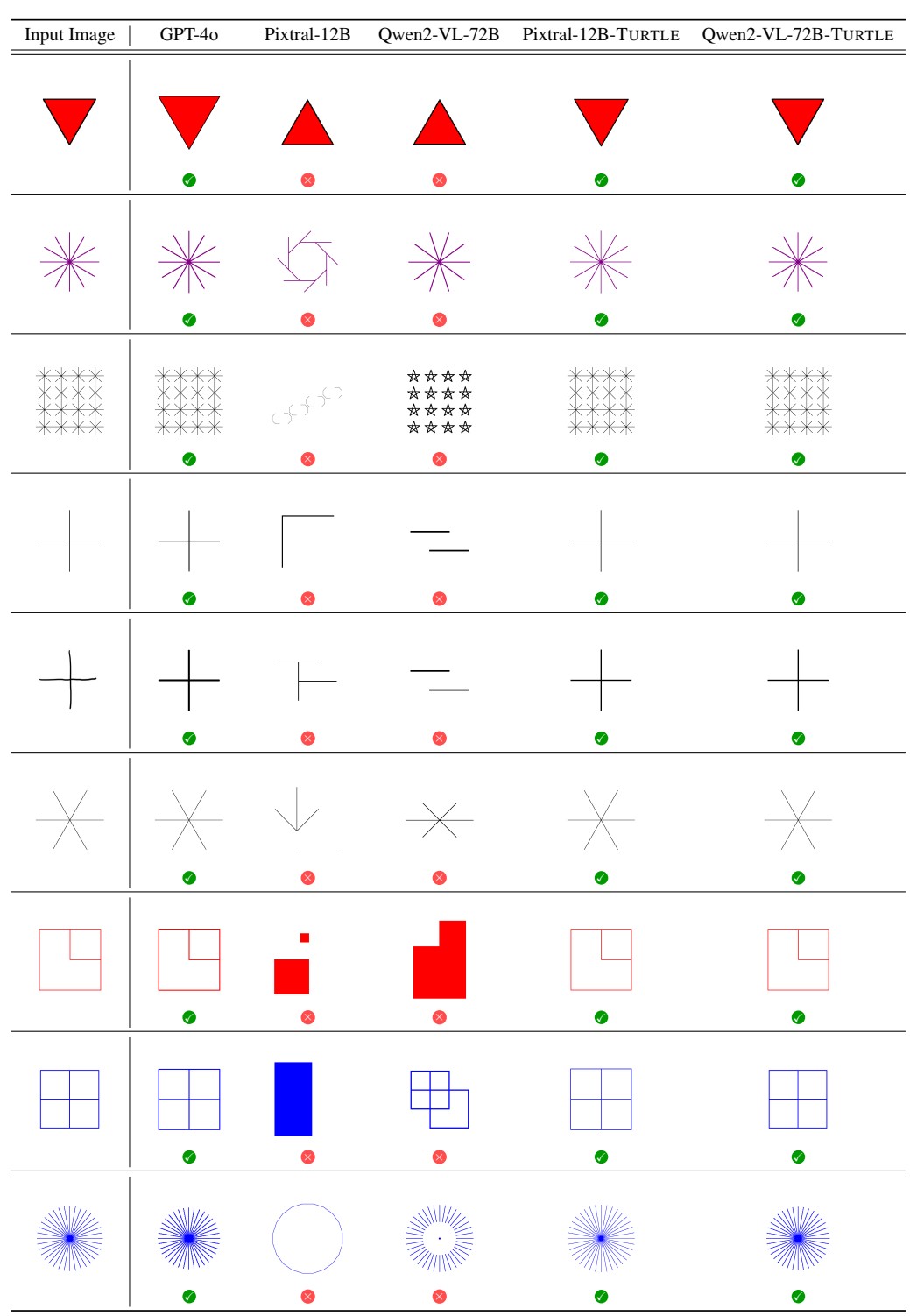

Figure 22: Example tasks that are successfully solved by GPT-4o and our fine-tuned models (i.e., Pixtral-12B-TURTLE, Qwen2-VL-72B-TURTLE), but not solved by the base models (Pixtral-12B, Qwen2-VL-72B) in the TURTLEAI-DS dataset. A total of 17 tasks (2.06%) match this criteria. Each row shows a ground truth image (leftmost) followed by the corresponding images generated by executing the each model's generated Python code. Success (✓) and failure (✗) are determined by our evaluation framework using symbolic comparison.

Figure 23: Tasks that are successfully solved by base models (i.e., GPT-4o, Pixtral-12B, Qwen2-VL-72B), but not solved by our fine-tuned models (i.e., Pixtral-12B-TURTLE, Qwen2-VL-72B-TURTLE) in the TURTLEAI-DS dataset. A total of 3 tasks (0.36%) match this criteria. Each row shows a ground truth image (leftmost) followed by the corresponding images generated by executing the each model's generated Python code. Success (✓) and failure (✗) are determined by our evaluation framework using symbolic comparison.

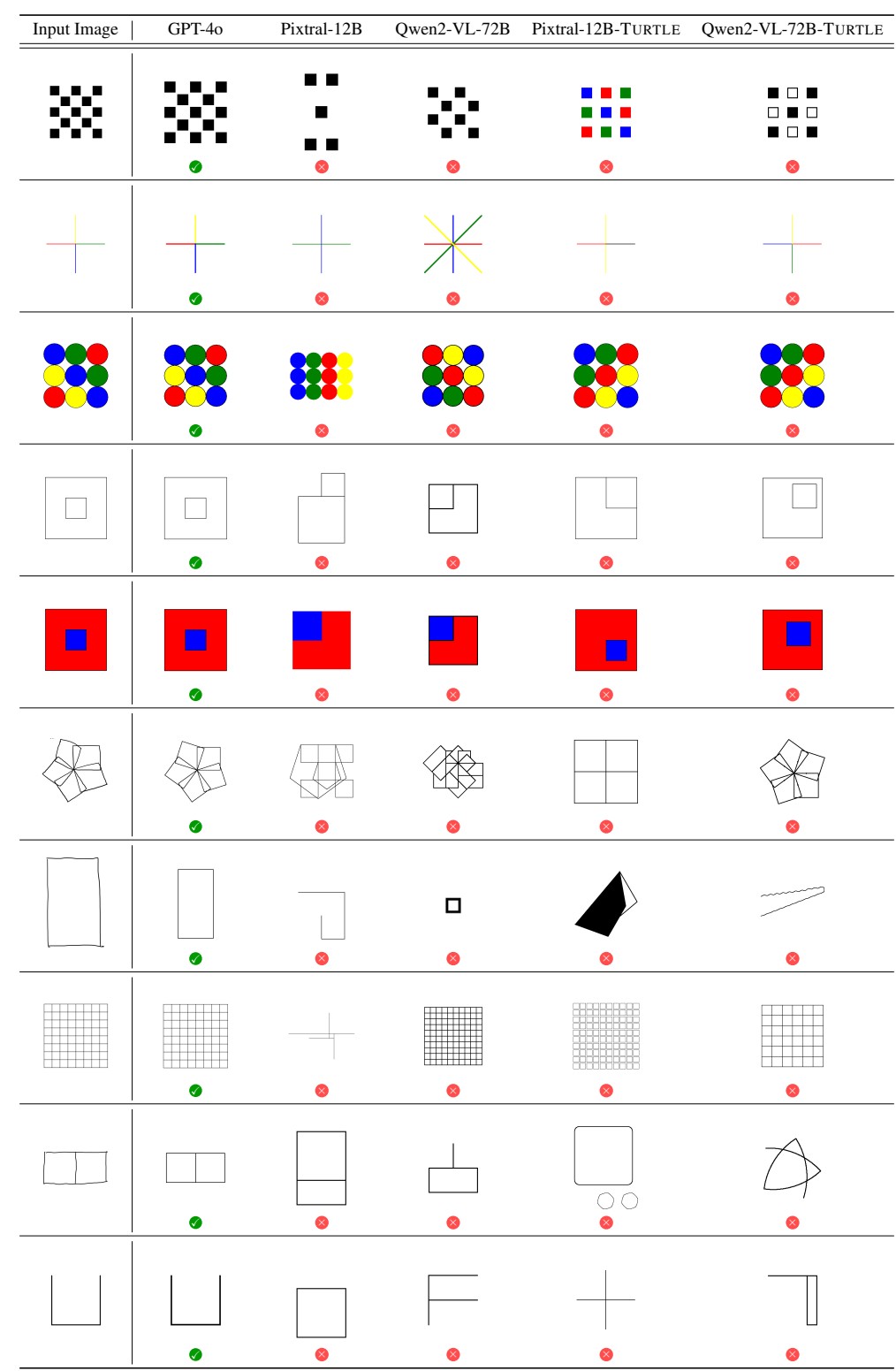

Figure 24: Example tasks that are only successfully solved by GPT-4o and not by Pixtral-12B, Qwen2-VL-72B, Pixtral-12B-TURTLE, or Qwen2-VL-72B-TURTLE models in the TURTLEAI-DS dataset. A total of 23 tasks (2.79%) match this criterion. Each row shows a ground truth image (leftmost) followed by the corresponding images generated by executing the each model's generated Python code. Success (✓) and failure (✗) are determined by our evaluation framework using symbolic comparison.

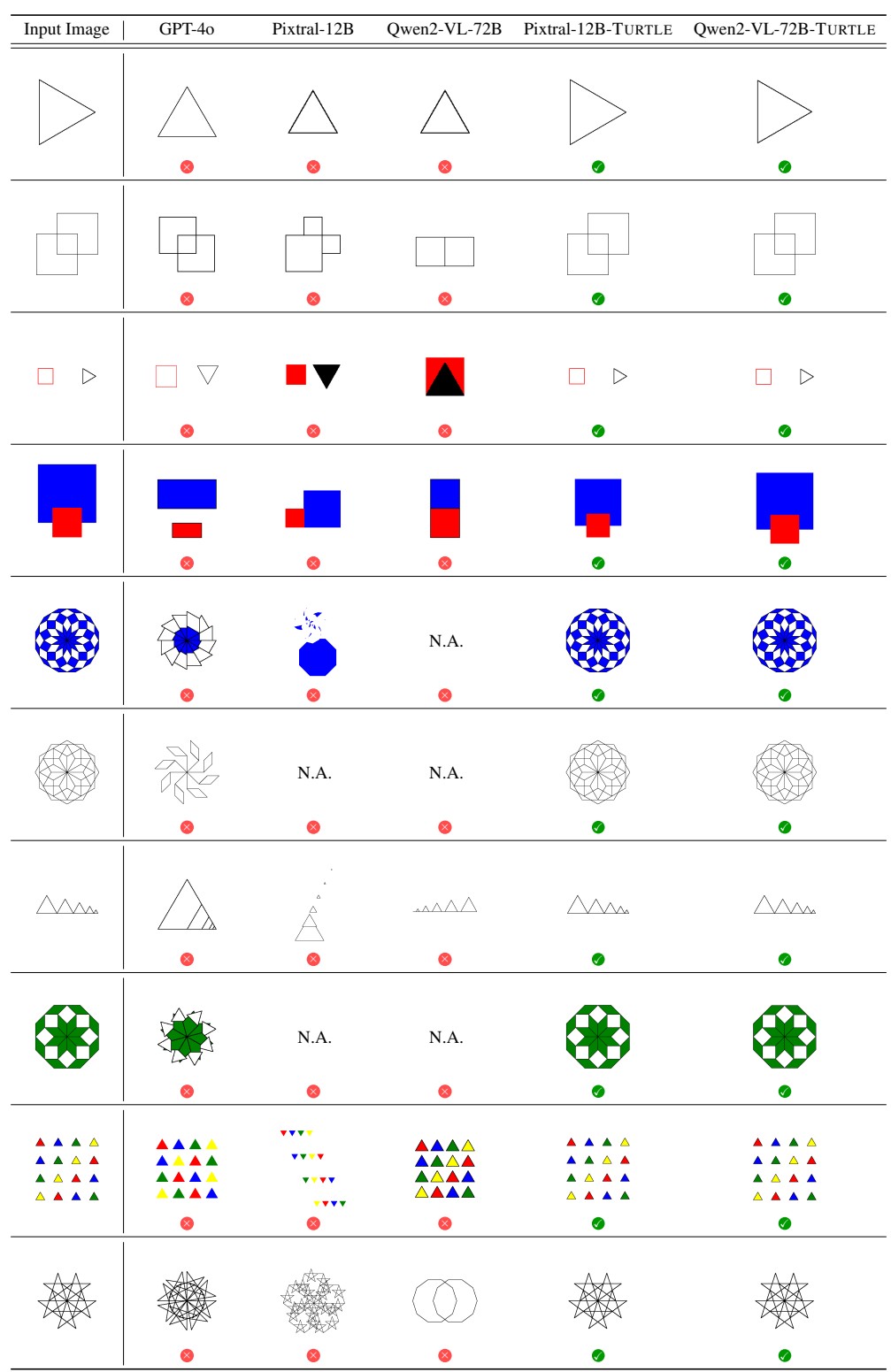

Figure 25: Example tasks that are only successfully solved by our fine-tuned models (i.e., Pixtral-12B-TURTLE, Qwen2-VL-72B-TURTLE) and not by Pixtral-12B, Qwen2-VL-72B, or GPT-4o models in the TURTLEAI-DS dataset. A total of 54 tasks (6.56%) match this criterion. Each row shows a ground truth image (leftmost) followed by the corresponding images generated by executing the each model's generated Python code. Success (✓) and failure (✗) are determined by our evaluation framework using symbolic comparison.

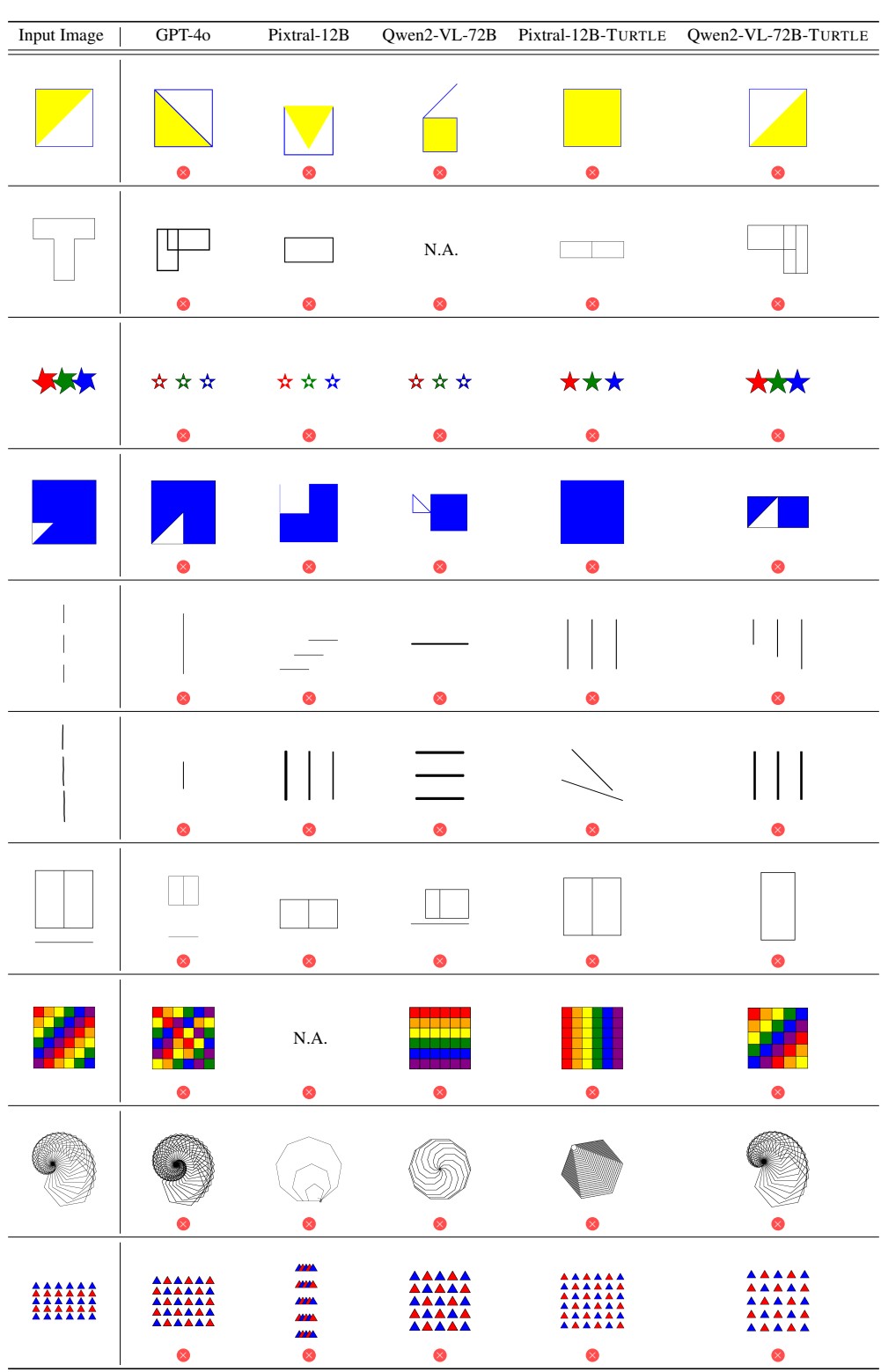

Figure 26: Tasks that are not solved by any of the models (i.e., GPT-4o, Pixtral-12B, Qwen2-VL-72B, Pixtral-12B-TURTLE, Qwen2-VL-72B-TURTLE) in the TURTLEAI-DS dataset. A total of 591 tasks (71.81%) match this criteria. Each row shows a ground truth image (leftmost) followed by the corresponding images generated by executing the each model's generated Python code. Success (✓) and failure (✗) are determined by our evaluation framework using symbolic comparison.

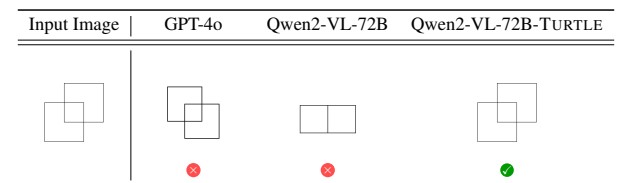

| Input Image | GPT-4o | Qwen2-VL-72B | Qwen2-VL-72B-Turtle |

(a) Input images and the generated images by different models.

```
Here is the Python code to
replicate the given geometric
pattern using the Turtle
Graphics library. The pattern
consists of two overlapping
squares.

```python
def draw(t):
    """Draw two overlapping
    squares."""
    def draw_square(t, size):
        """Helper function to draw
        a square of a given
        size."""
        for _ in range(4):
            t.forward(size)
            t.right(90)

    # Set up the turtle
    t.penup()
    t.goto(-50, 50) # Move to the
    starting position for the
    first square
    t.pendown()

    # Draw the first square
    draw_square(t, 100)

    # Move to the starting
    position for the second square
    t.penup()
    t.goto(0, 0)
    t.pendown()
    # Draw the second square
    draw_square(t, 100)
```

### Explanation:
1. **`draw_square` Function**:
   - A helper function to draw a
   square of a given size.
   - The turtle moves forward by
   the specified size and turns
   90 degrees to the right four
   times to complete the square.

2. **Positioning**:
   - The turtle starts at `(-50,
   50)` to draw the first square.
   - After completing the first
   square, the turtle moves to
   `(0, 0)` to draw the second
   square, ensuring the overlap.

3. **Overlapping**:
   - The two squares overlap at
   their bottom-left and
   top-right corners, as shown
   in the image.

### Usage:
To execute the code, use the
following:
```python
import turtle
# Create a turtle object
t = turtle.Turtle()
# Call the draw function
draw(t)
# Keep the window open
turtle.done()
```
```

```
```python
import turtle

def draw(t):
    """Draw the specified geometric
    pattern."""
    # Set the turtle's speed
    t.speed(0)

    # Define the side length of the
    squares
    side_length = 100

    # Draw the first square
    t.penup()
    t.goto(-side_length / 2,
    -side_length / 2)
    t.pendown()
    for _ in range(4):
        t.forward(side_length)
        t.right(90)

    # Draw the second square
    t.penup()
    t.goto(side_length / 2,
    -side_length / 2)
    t.pendown()
    for _ in range(4):
        t.forward(side_length)
        t.right(90)

# Create a turtle object
t = turtle.Turtle()
draw(t)

# Keep the window open
turtle.done()
```

This code defines a function
`draw(t)` that uses a turtle
object `t` to draw the specified
geometric pattern. The pattern
consists of two overlapping
squares, each with a side length
of 100 units. The turtle starts at
the center of the screen and moves
to the starting position of each
square before drawing it. The
`turtle.done()` function is used
to keep the window open after
drawing is complete.
```

```
## Image Description

The image displays two overlapping
squares. The squares are of equal
size and are positioned such that one
square is shifted diagonally relative
to the other. The overlapping area
forms a smaller square in the center,
while the non-overlapping parts of
each square extend outward, creating
a visually symmetrical pattern. The
lines are black, and the background
is white.

## Analysis & Solution Code

To create the pattern shown in the
image using Turtle Graphics, we need
to draw two squares of the same size.
The first square is drawn starting
from the initial position. After
completing the first square, the
turtle is moved diagonally to a new
position to draw the second square.
This diagonal movement ensures that
the two squares overlap, creating the
desired pattern.

Here is the code with comprehensive
docstrings and comments to create the
pattern:

```python
def draw(t):
    """
    Draws two overlapping squares of
    the same size (200 units) with a
    diagonal offset.

    Args:
        t: Turtle graphics object
    """
    def draw_square(size):
        """
        Draws a square with the given
        side length.

        Args:
            size: The length of each
            side of the square.
        """
        for _ in range(4):
            t.forward(size) # Move the
            turtle forward by the side
            length
            t.left(90) # Turn the turtle
            left by 90 degrees to form a
            corner

    # Draw the first square
    draw_square(200)
    # Move the turtle diagonally to
    the starting position of the
    second square
    t.penup() # Lift the pen to move
    without drawing
    t.goto(100, 100) # Move to the new
    starting position
    t.pendown() # Put the pen down to
    start drawing again
    # Draw the second square
    draw_square(200)
```
```

| (b) GPT-4o | (c) Qwen2-VL-72B | (d) Qwen2-VL-72B-Turtle |

Figure 27: Example outputs generated by GPT-4o, Qwen2-VL-72B, and Qwen2-VL-72B-Turtle. (a) shows the input image and the generated images by executing the code generated by models. (b-d) show the outputs generated by different models.

| Input Image | GPT-4o | Qwen2-VL-72B | Qwen2-VL-72B-TURTLE |

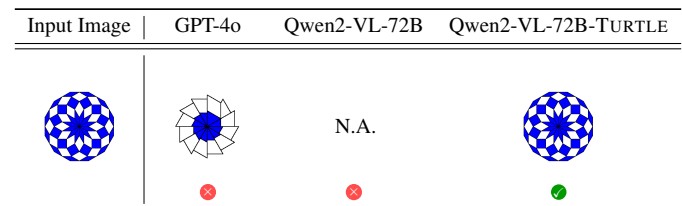

N.A.

(a) Input images and the generated images by different models.

```
To replicate the geometric pattern shown in the image
using Python's Turtle Graphics library, we need to
break the pattern into smaller components. The
pattern consists of a circular arrangement of blue
and white diamond shapes, forming a star-like
structure.

Here is the Python code to recreate the pattern:

```python
def draw(t):
    import turtle
    import math

    # Helper function to draw a diamond shape
    def draw_diamond(t, size, color):
        t.fillcolor(color)
        t.begin_fill()
        for _ in range(2):
            t.forward(size)
            t.left(60)
            t.forward(size)
            t.left(120)
        t.end_fill()

    # Helper function to draw a single segment of the
    pattern
    def draw_segment(t, size):
        # Draw the blue diamond
        draw_diamond(t, size, "blue")
        # Position for the white diamond
        t.forward(size)
        t.left(60)
        # Draw the white diamond
        draw_diamond(t, size, "white")
        # Return to the center
        t.right(60)
        t.backward(size)

    # Main function to draw the full pattern
    def draw_pattern(t, size, segments):
        for _ in range(segments):
            draw_segment(t, size)
            t.right(360 / segments)

    # Set up the turtle
    t.speed(0)
    t.penup()
    t.goto(0, 0)
    t.pendown()

    # Draw the pattern
    draw_pattern(t, 50, 12) # Adjust size and number
    of segments as needed

    # Hide the turtle
    t.hideturtle()
```

### Explanation:

<some text is omitted here to save space>

### Usage:

<some text is omitted here to save space>

This code will produce the desired geometric pattern.
Adjust the 'size' and 'segments' parameters in the
'draw_pattern' function to modify the pattern's
appearance.
```

```
## Image Description

The image displays a symmetrical, flower-like pattern
composed of geometric shapes. The primary shape is a
dodecagon (a 12-sided polygon) repeated in a circular
arrangement. Each dodecagon is filled with a blue
color, and the spaces between the dodecagons are
white, creating a checkerboard-like effect. The
pattern exhibits rotational symmetry, with each
dodecagon rotated slightly relative to its neighbors,
forming a visually appealing star-like structure at
the center. The overall design is intricate yet
balanced, with the blue and white sections
contrasting sharply against each other.

## Analysis & Solution Code

To create the pattern shown in the image using Turtle
Graphics, we need to follow these steps:

1. **Set Up the Turtle**: Initialize the turtle
graphics and set the fill color to blue.
2. **Define the Dodecagon**: Create a function to
draw a single dodecagon. A dodecagon has 12 sides,
and each internal angle is 150 degrees (calculated as
\( \frac{180 \times (12-2)}{12} \)).
3. **Draw the Pattern**: Use a loop to draw 12
dodecagons, each rotated by 30 degrees (360 degrees /
12) relative to the previous one. This rotation
ensures the symmetrical arrangement of the dodecagons.
4. **Fill the Shape**: Use the `begin_fill` and
`end_fill` methods to fill the dodecagons with the
specified color.

Here is the code with comprehensive docstrings and
comments to create the pattern:

```python
def draw(t):
    """
    Draws a symmetrical, flower-like pattern composed
    of 12 blue-filled dodecagons.
    Each dodecagon is rotated by 30 degrees relative
    to the previous one, creating a
    visually appealing star-like structure at the
    center.

    Args:
        t: Turtle graphics object
    """
    t.fillcolor('blue') # Set the fill color to blue

    def draw_dodecagon():
        """
        Draws a single dodecagon with 12 sides.
        Each side is 50 units long, and the internal
        angle is 150 degrees.
        """
        for _ in range(12):
            t.forward(50) # Move the turtle forward by
            50 units
            t.left(30) # Turn the turtle left by 30
            degrees

    t.begin_fill() # Start filling the shape
    for _ in range(12):
        draw_dodecagon() # Draw a dodecagon
        t.left(30) # Rotate the turtle by 30 degrees
        for the next dodecagon
    t.end_fill() # End filling the shape
```
```

(b) GPT-4o                    (c) Qwen2-VL-72B-TURTLE

Figure 28: Example outputs generated by GPT-4o and Qwen2-VL-72B-TURTLE. (a) shows the input image and the generated images by executing the code generated by models. (b) and (c) show the outputs generated by different models. Qwen2-VL-72B's output is not shown since it generates repetitive text.

2052
2053
2054
2055
2056
2057
2058
2059
2060
2061
2062
2063
2064
2065
2066
2067
2068
2069
2070
2071
2072
2073
2074
2075
2076
2077
2078
2079
2080
2081
2082
2083
2084
2085
2086
2087
2088
2089
2090
2091
2092
2093
2094
2095
2096
2097
2098
2099
2100
2101
2102
2103
2104
2105

## Prompt for Generating Code from Image

**<image>** You are a Turtle Graphics programmer tasked with creating Python code to replicate a specified geometric pattern using the Turtle Graphics library.

### Task Overview:

Analyze the provided image of a geometric pattern. Carefully break down the pattern into individual shapes, colors, angles, and layout components. Using this information, write Python code within a function called `draw(t)`, where `t` is a Turtle object. Assume:
- The turtle starts at the center of the screen at coordinates `(0, 0)`.
- The turtle initially faces east (to the right).

The goal is for your `draw(t)` function to accurately recreate the pattern shown in the image, including its positioning, angles, colors, and details.

### Requirements:

1. **Code Structure:**
    - Place all code inside the `draw(t)` function.
    - The function takes a turtle object `t` as input.
    - Format your code using triple backticks with the 'python' language specifier, i.e., ```python```.

    Example format:
    ```python
    def draw(t):
        # Your code here
    ```
2. **Color Accuracy:**
    - Match colors in the image exactly, both for fills and outlines.
3. **Pattern Precision:**
    - Reproduce the pattern as accurately as possible, maintaining symmetry, shapes, and angles.
4. **Self-Contained:**
    - Do not include code outside the `draw(t)` function.
    - All necessary imports, variables, and helper functions should be inside `draw(t)`.

### Execution Context:

Your `draw(t)` function will be called in the following manner:

```python
import turtle

def draw(t):
    # Describe the drawing steps here
    pass

t = turtle.Turtle()
draw(t)
```

### Example Outputs:

- **Example 1 – Drawing a Rectangle:**
```python
def draw(t):
    """Draw a rectangle."""
    def draw_rectangle(t):
        # Draw a rectangle with side length 10
        for _ in range(4):
            t.forward(10)
            t.right(90)
    draw_rectangle(t)
```

- **Example 2 – Drawing a circle:**
```python
def draw(t):
    """Draw a circle."""
    import math

    def draw_circle(t, radius):
        circumference = 2 * math.pi * radius
        step_length = circumference / 360
        step_angle = 1

        for _ in range(360):
            t.forward(step_length)
            t.left(step_angle)

    draw_circle(t, 50)
```

*Note:* The examples are simplified. Your final code may require nested loops or additional logic to fully replicate complex patterns.

Now, write the code for the `draw(t)` function to recreate the pattern shown in the image as closely as possible in terms of shape, color, and structure.

Figure 29: Prompt template for code synthesis from visual input.

2106
2107
2108
2109
2110
2111
2112
2113
2114
2115
2116
2117
2118
2119
2120
2121
2122
2123
2124
2125
2126
2127
2128
2129
2130
2131
2132
2133
2134
2135
2136
2137
2138
2139
2140
2141
2142
2143
2144
2145
2146
2147
2148
2149
2150
2151
2152
2153
2154
2155
2156
2157
2158
2159

## Prompt for the Reference-guided Code Mutation Stage of TURTLEAI-Datagen

You are a turtle graphics programmer tasked with **analyzing and applying code adaptations** in Python using the Turtle Graphics library. You are given **two reference codes** that perform a certain drawing task. Your mission is to:

1. **Identify how the adaptation is done** from the first code to the second code.
2. **Summarize the adaptation** in a **high-level way**, so it can be applied to any other code.
3. **Apply the core idea of the adaptation** to a new piece of code provided.

### Key Requirements for Code Adaptation:

1. Syntactic Correctness:
   - The adapted code must be **syntactically correct** and free of errors.

2. Structural and Logical Consistency:
   - Maintain the **structural integrity** and **logical flow** of the original code.
   - Ensure that no unintended behavior is introduced by the adaptation.

3. Geometric Structure & Symmetry (if applicable):
   - Ensure that all drawings consist of **clear geometric shapes** with **symmetry** and **geometric accuracy**.

4. Visual Clarity & Simplicity:
   - The output should be **visually clear** and **simple**.
   - Avoid overly complex designs that may confuse or clutter the output.

5. Function and Code Requirements:
   - Define the function `draw(t)` that contains all the drawing code.
   - Use appropriate Turtle Graphics library commands within the `draw(t)` function.
   - Only provide the `draw(t)` function. **Do not include import statements** or other code outside of the `draw()` function.

6. Different Output:
   - The **adapted code must generate a different drawing** compared to the original new code.
   - The drawing must be a different shape or have a distinct pattern to clearly show the adaptation's impact.

### Your Task:
**Reference Code 1:**
```python
{reference_code_1}
```

**Reference Code 2:**
```python
{reference_code_2}
```

**New Code to Adapt:**
```python
{code_to_adapt}
```

Now, follow these steps:

1. Analyze the Adaptation:
   - Examine how **Reference Code 1** is adapted into **Reference Code 2**.
   - Summarize the adaptation in a **high-level way** that can be applied to other codes.

2. Apply the Adaptation:
   - Apply the core idea of the adaptation to the **New Code to Adapt**.
   - Provide the **Adapted Code** that reflects this adaptation.
   - Ensure the adapted code is **syntactically correct** and that the resulting drawing after execution meets all the specified requirements (geometric structure, symmetry, visual clarity, simplicity, etc.).

**Adapted Code:**

Provide your adapted code here. Ensure it meets all the specified requirements, especially that it must generate a different drawing compared to the original new code. Use the following Python code block format:

```python
def draw(t):
    # Your adapted code here
```

Figure 30: Prompt template for reference-guided code generation.

---

**Prompt for the Elite Selection Stage of TURTLEAI-Datagen**

**<image>** You are an evaluator responsible for automatically assessing the quality of a turtle graphics programming task using the following rubrics. Each rubric evaluates different aspects of the task, including its clarity, difficulty, alignment with programming concepts, and creativity. Please assign a score from 0 to 10 for each rubric and provide an explanation for your scoring. Each rubric has equal weight, and the rubrics are as follows:

### Rubrics Breakdown:

1. **Geometric Structure & Symmetry**
- Score Breakdown (0-10):
  - 9-10: Perfect geometric accuracy and symmetry – all elements are precisely aligned and balanced.
  - 6-8: Mostly symmetric with minor imperfections – slight deviations that do not detract from overall symmetry.
  - 3-5: Some geometric or symmetry issues – noticeable asymmetries or inaccuracies in shape.
  - 0-2: Significant asymmetry and inaccuracies – major deviations from expected geometric forms.
2. **Visual Appeal, Clarity & Simplicity**
- Score Breakdown (0-10):
  - 9-10: Clear, simple design with purposeful aesthetics – easily understood and visually harmonious. No unnecessary complexity.
  - 6-8: Generally clear design with good balance, but has minor complexity or visual elements that could be simplified.
  - 3-5: Either overly complex, lacks visual harmony, or has clarity issues – may have unnecessary elements or confusing design choices.
  - 0-2: Significant issues with clarity or complexity – cluttered, difficult to interpret, or contains many unnecessary elements.
3. **Structural Coherence**
- Score Breakdown (0-10):
  - 9-10: Strong structural integrity – design is cohesive, whether through repeated patterns, basic geometric shapes, or a purposeful unique design.
  - 6-8: Good structure with minor imperfections – mostly coherent with slight inconsistencies.
  - 3-5: Basic structure present but with noticeable flaws – some elements may seem out of place or poorly integrated.
  - 0-2: Weak or unclear structure – lacks a clear organizational pattern or design logic.
4. **Alignment & Positioning**
- Score Breakdown (0-10):
  - 9-10: Excellent alignment and positioning – all elements are precisely placed and aligned.
  - 6-8: Good alignment with minor issues – generally well-positioned with slight misalignments.
  - 3-5: Some misalignment – noticeable but not critical positioning errors.
  - 0-2: Noticeable misalignment – significant positioning errors that affect the overall design.
5. **Educational Value & Solvability**
- Score Breakdown (0-10):
  - 9-10: Excellent educational value – pattern complexity is appropriate for learning, clear objectives, and perfectly balanced difficulty that students can reasonably solve.
  - 6-8: Strong educational value – complexity is manageable for students with some guidance, mostly clear and appropriately challenging without being overwhelming.
  - 3-5: Moderate educational value – either too simple to be educational or too complex for students to reasonably solve. Would require significant modifications to be classroom-ready.
  - 0-2: Poor educational value – not suitable for classroom use due to excessive complexity, confusing structure, or contains sensitive/inappropriate imagery (e.g., Swastika, Confederate flag, etc.).
6. **Color Usage & Necessity**
- Score Breakdown (0-10):
  - 9-10: Excellent use of minimal colors – either black & white only, or uses very few colors (<5) with clear purpose that enhances understanding.
  - 6-8: Acceptable color usage – slightly more colors than necessary but not distracting. Could be simplified without losing meaning.
  - 3-5: Problematic color usage – too many colors or colors used without clear purpose. Would be clearer with fewer colors.
  - 0-2: Poor color usage – excessive number of colors, random color choices, or colors that make the pattern harder to understand.

### Final Evaluation Instructions:

Once you have evaluated each category and assigned scores, sum up all the individual rubric scores. Since each rubric has equal weight, no additional multiplication is needed. The **final score** is simply the sum of all rubric scores. Summarize the individual scores and explanations, then provide the final score (out of 60).

### Expected JSON Output:

Please format the final evaluation as a JSON object using the following short keys:
- **geometry:** Geometric Structure & Symmetry
- **visual:** Visual Appeal & Clarity
- **structure:** Structural Coherence
- **alignment:** Alignment & Positioning
- **education:** Educational Value & Solvability
- **color:** Color Usage & Necessity
- **final_score:** Final score out of 60

### Example JSON Output:
```json
{
  "geometry": {"score": "<score>", "explanation": "<explanation>"},
  "visual": {"score": "<score>", "explanation": "<explanation>"},
  "structure": {"score": "<score>", "explanation": "<explanation>"},
  "alignment": {"score": "<score>", "explanation": "<explanation>"},
  "education": {"score": "<score>", "explanation": "<explanation>"},
  "color": {"score": "<score>", "explanation": "<explanation>"},
  "final_score": "<final_score>"
}
```
Now, evaluate the turtle graphics task based on the provided image. This image was created using turtle graphics. Please assess its quality using the rubrics outlined above, and provide the final evaluation in the JSON format shown in the example.

Figure 31: Prompt template for the elite selection stage in TURTLEAI-Datagen.



## Prompt for the CoT Labeling Stage of TURTLEAI-Datagen

**\<image\>** Your task is to optimize a provided Python code snippet that uses Turtle Graphics, ensuring it is minimal, cleanly documented, and fully aligned with the generated image output. You will be provided with:

1. A Python code snippet using Turtle Graphics.
2. The actual image output generated by this code.

## Your Responsibilities:

1. **Describe the Image**
   - Provide a detailed description of the visual pattern in the image **without referencing the code**, focusing on geometric shapes, symmetry, colors, and overall structure.

2. **Optimize the Code**
   - Identify and remove redundant code segments that do not contribute to the visual output, and simplify the logic to enhance readability, ensuring the
   **final output remains visually identical**.
   - After optimizing the code, provide a detailed step-by-step explanation of how the code generates the image, linking visual features to the corresponding steps.

3. **Add Documentation and Comments**
   - Add a descriptive docstring for the provided code snippet, explaining its purpose, parameters, and any outputs. Include clear and concise inline comments to make the code understandable.

## Formatting Instructions:
   - **Markdown-Only Response:** Format your entire response in markdown, enclosed in a single markdown block.
   - **Output Focus:** Provide only the optimized `draw(t)` function within the markdown block, excluding any setup or unrelated code.

## Provided Code Snippet

Below is the code snippet that generates the image you are analyzing:

```python
{code}
```

Please provide your response in the following markdown format:

```markdown
## Image Description

The image displays...

[Provide a detailed description of the visual pattern]

## Analysis & Solution Code

To create the pattern shown in the image using Turtle Graphics, we need to...
[Explain how to create this pattern using Turtle Graphics, describing the logical steps needed to reproduce the image]

Here is the code with comprehensive docstrings and comments to create the pattern:
```python
def draw(t):
    """
    [Function description]

    Args:
        t: Turtle graphics object
    """
    # Your simplified code with comments
```
```

**Important:** Write your response as if you are only looking at the image, without referencing any provided code (e.g., do not mention 'modified code', 'optimized code', or 'provided code' in your response inside the markdown block).



Figure 32: Prompt template for the CoT labeling stage in TURTLEAI-Datagen.

