# OpenReview forum: "TURTLEAI: Benchmarking Multimodal Models in Turtle Graphics for Visual Programming and Reasoning"
_ICLR.cc/2026/Conference — ICLR 2026 Conference Withdrawn Submission_

### Official Review · Reviewer_FsFK · 2025-10-31

**Soundness:** 3
**Presentation:** 3
**Contribution:** 2
**Rating:** 4
**Confidence:** 4

**Summary:**

The paper proposes a multimodal benchmark, TURTLEAI, which challenges vision-language models (VLMs) to generate Python code for drawing patterns. TURTLEAI consists of three components: TURTLEAI-DS (a collection of datasets), TURTLEAI-Eval (an evaluation framework), and TURTLEAI-Datagen (a data generation framework).
TURTLEAI-DS contains pairs of images and their corresponding Python code. A VLM is required to generate Python code based on a given image. The generated image is then compared with the original image using TURTLEAI-Eval. TURTLEAI-Datagen is designed to generate (image, code) pairs and chain-of-thought (CoT) reasoning examples for fine-tuning VLMs.
Experimental results demonstrate that existing VLMs struggle to perform well on these tasks, but fine-tuning the models significantly improves their performance.

**Strengths:**

-	The paper introduces a new benchmark to facilitate the development of VLMs.
-	It reveals that existing VLMs struggle with visual programming tasks.
-	It proposes a data generation framework for fine-tuning.

**Weaknesses:**

- In TURTLEAI-Eval, which compares drawings in a transformed, normalized space, the line width is standardized to a fixed value of 1. However, certain shapes may require width information for accurate representation. For instance, a solid rectangle can be considered as a very thick line, and its width plays a crucial role in distinguishing it from other shapes.
- In Symbolic comparison and Embedding-based comparison, there is a predefined threshold. What impact does this threshold have on the evaluation results? How is this threshold determined, and is there any experimental evidence to support it?
- In TURTLEAI-DATAGEN, two codes are randomly selected from the dataset to extract their high-level mutation pattern. However, the selected codes may not have a clear mutation pattern (e.g., adding a loop to the code).
- In stage 3 of TURTLEAI-DATAGEN, is there any quality check for the generated CoT reasoning?

**Questions:**

If the writing could clarify the code mutation process in more detail, especially the extraction of high-level mutation patterns from reference codes, and include some concrete examples, it would enhance the quality of the paper.

---

> ### Author Response · Authors · 2025-11-26
>
> Dear Reviewer,
>
> We sincerely appreciate your thoughtful and meticulous review. Below, we address your points.
>
> ---
>
> > W1: In TURTLEAI-Eval, which compares drawings in a transformed, normalized space, the line width is standardized to a fixed value of 1. However, ...
>
> We agree that drawing a solid rectangle as an extremely thick line is a valid but creative/tricky solution. In our current pipeline, line width is normalized before comparison, so such a program would indeed be judged incorrect against a ground-truth filled rectangle. This is a corner case that our evaluation cannot capture.
>
> We believe this reflects a trade-off in the evaluation design. With normalization, we can use symbolic comparison, which we show in Appendix D to be more accurate and stable than embedding-based comparison. If we removed normalization, symbolic comparison may not work well; we would have to rely on embedding-based comparison in the original space, which could recognize the thick-line rectangle as close to the filled rectangle, but is overall less precise and introduces more noise.
>
> In our benchmark, we prioritize accuracy and therefore adopt a normalized evaluation setting. We had discussed such limitation and possible directions for future work (Section 6) to account for additional details such as size, position, and line width. We hope this could clarify your concern.
>
> ---
>
> > W2: In Symbolic comparison and Embedding-based comparison, there is a predefined threshold. What impact does this threshold have on the evaluation results? How is this threshold determined, and is there any experimental evidence to support it?
>
> The threshold value of 0.95 was chosen based on comparison with human-annotated ground truth (see **Appendix D**).
>
> Specifically, we first chose a well-performing base model (i.e., GPT-4o). Then we manually annotated the GPT-4o outputs on the evaluation dataset by labeling each result as a success or failure. We then evaluated the alignment between these manual annotations and the outcomes from our symbolic comparison under different threshold values. Through this process, we found that a threshold of 0.95 provided the best alignment with human annotation, achieving a precision of 0.974, recall of 0.937, F1 score of 0.955, and accuracy of 0.991. These metrics suggest that 0.95 is a highly effective threshold for symbolic comparison.
>
> For the embedding-based comparison, we conducted a similar process by using different thresholds and measuring performance against the human annotations. We found that a threshold of 0.95 yielded the best F1 score of 0.896. Please refer to Figure 17 in Appendix D for a detailed visualization of the results.
>
> ---
>
>
> > W3: In TURTLEAI-DATAGEN, two codes are randomly selected from the dataset to extract their high-level mutation pattern. However, the selected codes may not have a clear mutation pattern (e.g., adding a loop to the code).
>
> We agree that two randomly selected codes may not always exhibit a clear high-level mutation pattern. This is not a problem for our pipeline and is, in fact, part of the design.
>
> In TURTLEAI-DATAGEN, the selected pair is used only as **inspiration** for mutation. When a clear transformation exists, the model can extract it (e.g., rotation, scaling, adding repetition). When no meaningful pattern exists, three outcomes are possible, all acceptable:
>
> 1. **The model outputs identical code.** These cases are removed automatically during deduplication stage.
> 2. **The model outputs a hallucinated but valid and meaningful mutated code.** This is acceptable, and even beneficial. This is because unexpected mutations can increase diversity of the generated code.
> 3. **The model outputs hallucinated low-quality code.** These cases are filtered out during the elite-selection stage, which keeps only valid and high-quality samples.
>
> Thus, the synthesis process does not rely on every randomly selected code pair having a clear pattern.
>
> I hope this clarifies your concern.
>
> ---
>
> > W4: In stage 3 of TURTLEAI-DATAGEN, is there any quality check for the generated CoT reasoning?
>
> We do not quality-check the generated CoT reasoning in the data synthesis pipeline. CoT is only used as an auxiliary training signal, while evaluation depends solely on the final code.
>
> ---
>
> > Q1: If the writing could clarify the code mutation process in more detail, especially the extraction of high-level mutation patterns from reference codes, and include some concrete examples, it would enhance the quality of the paper.
>
> Thank you for the suggestion. We had provided examples and additional explanations of the mutation process in **Appendix E.1**. Due to space constraints, we could not include detailed examples in the main paper.
>
> ---
>
> We hope these responses address your concerns. We appreciate your detailed feedback and will incorporate these suggestions into our revision.

---

### Official Review · Reviewer_Ecws · 2025-11-01

**Soundness:** 1
**Presentation:** 2
**Contribution:** 2
**Rating:** 2
**Confidence:** 3

**Summary:**

The paper introduces TURTLEAI, a dataset and benchmark for visual to code generation in Turtle Graphics, along with a synthetic data pipeline that reportedly improves model performance.      However, the benchmark is evaluated only on its own synthetic distribution, and the observed improvements are likely due to domain exposure rather than genuine reasoning gains.      OOD results show that finetuning actually makes models worse on hand-drawn sketches, suggesting harmful overfitting.      No external datasets, no comparisons to existing sketch to code benchmarks, outdated baselines, and poor presentation quality further weaken the contribution.      While releasing the dataset could be valuable to the community, the scientific impact and novelty of the paper are limited.

**Strengths:**

1. Clear problem formulation and a well defined task scope.
The paper focuses on visual-to-code generation via Turtle Graphics, which is a structured setting where correctness can be objectively evaluated via execution.

2.  Infrastructure contribution.
The authors provide a dataset, an evaluator, and code for data synthesis and benchmarking.      If fully released, the benchmark could serve as a reproducible testbed for small scale visual to program induction.

3.  Diagnostic experiments.
The paper analyzes failure types and includes limited OOD testing, which helps reveal the limitations of current VLMs on synthetic geometric tasks.

**Weaknesses:**

1.   Major validity issue: the benchmark is only evaluated on its own synthetic data, with no external datasets or established baselines.
The paper trains on data generated by its own pipeline and then evaluates on a benchmark created from the same distribution.     This closed loop validation prevents demonstrating scientific novelty, generality, or impact.     There is no evidence that performance gains reflect reasoning improvements rather than domain overfitting or memorization.

2.  OOD experiments contradict the core claims.
In Figure 7, models fine-tuned on TURTLEAI data perform significantly worse on hand-drawn OOD sketches.     Since both tasks are visually to code mapping, this drop indicates the model becomes *less* general and more brittle after training suggesting harmful domain overfitting rather than capability improvement.     This undermines the core contribution of the dataset and training pipeline.

3.  The baseline success rates are extremely low (~10%), making “20% improvement” uninformative.
Because current VLMs have never been trained on synthetic Turtle-style graphics, any improvement after domain exposure is expected and does not imply methodological novelty.     The paper does not demonstrate that its data synthesis strategy is better than simpler alternatives such as random sampling, Self-Instruct, Evol-Instruct, or manual template expansion.

4.  No comparison with existing visual to program or sketch-to-code benchmarks.
The work ignores relevant prior datasets (e.g., SVG/TikZ program induction, sketch-to-code datasets, visual UI-to-code tasks).     Without external validation, it is unclear whether TURTLEAI measures general reasoning or just fits a narrow toy domain.

5.  The data and task space are extremely toy-like.
All visuals are synthetic geometric primitives with perfect rendering (no noise, occlusion, perspective, thickness variation).     Claims about “general visual reasoning” or “broad program synthesis” are overstated relative to the simplicity of the domain.

6.  Paper presentation quality is below conference standard.
All tables are mislabeled as figures, and captions are consistently placed incorrectly.     This violates standard formatting guidelines and indicates insufficient care in preparation.

**Questions:**

1.  Why are there no experiments on existing sketch-to-code or visual program induction benchmarks?     Without external evaluation, how can the benchmark claim scientific relevance beyond its own synthetic sandbox?
2.  In Fig.7, why does fine-tuning severely degrade performance on hand-drawn sketches?     Doesn’t this imply the dataset harms general visual programming rather than improves it?
3. What proportion of generated samples are incorrect, redundant, or semantically invalid?     Is there any human auditing, or are models only self-evaluating their own output?
4.  How do you demonstrate that the proposed data synthesis approach is superior to simpler strategies (e.g., template mutation or random augmentation)?
5. Many baselines used (e.g., GPT-4V, Qwen2-VL) are outdated relative to 2024–2025 VLMs.     Why are recent models (GPT-4o, GPT-5，Qwen2.5-VL，Qwen3-VL，etc.) missing?

---

> ### Author Response · Authors · 2025-11-26
> **Rebuttal (1/2)**
>
> Dear Reviewer,
>
> Thank you for taking the time to provide such detailed feedback. We would like to address each of your concerns in detail below.
>
> ---
>
> > W1. Major validity issue: the benchmark is only evaluated on its own synthetic data, with no external datasets or established baselines. The paper trains on data generated by its own pipeline and then evaluates on a benchmark created from the same distribution...
>
> Our benchmark is **not** only evaluated on its own synthetic data.
>
> As described in Section 3.2 (lines 191–200), TURTLEAI-DS consists of three datasets:
> - **DSBasic**: this is a set of **real-world tasks** curated from the visual programming platform XLogoOnline. These tasks are designed by domain experts and used by real students; they are **not** generated by our synthesis pipeline.
> - **DSCraft**: this is a **manually hand-drawn dataset**, created by a human who redraws each DSBasic task on a tablet. This dataset serves as an in-domain OOD dataset, and our fine-tuned models are never trained on any hand-drawn sketches.
> - **DSSyn**: this is a **synthetic dataset** generated by using DSBasic as the seed dataset for data generation, followed by manual quality filtering and selection.
>
>
> Overall, two of the three evaluation datasets (DSBasic and DSCraft) are **human-created** and not generated by our data generation framework. And DSCraft is deliberately curated to be used as an in-domain OOD dataset. We hope this clarifies the misunderstanding.
>
>
>
> ---
>
> > W2. OOD experiments contradict the core claims. In Figure 7, models fine-tuned on TURTLEAI data perform significantly worse on hand-drawn OOD sketches...
>
>
> The performance drop on hand-drawn sketches is **not caused by fine-tuning**. The base models already struggle significantly on DSCraft due to the large visual style shift introduced by human drawings. This is visible in Fig. 7: DSCraft accuracy is low before any fine-tuning.
>
> In fact, fine-tuning **does not harm** generalization to this in-domain OOD setting. As shown in Fig. 7a, Qwen2-VL-72B's performance on DSCraft remains roughly stable after fine-tuning, and for weaker models (Qwen2-VL-7B, Pixtral-12B), fine-tuning even improves DSCraft accuracy. Moreover, our ablation in Fig. 7a shows that including CoT stage preserves or improves DSCraft performance. This indicates that the training pipeline does not introduce harmful overfitting to synthetic data; rather, with CoT labeling it can even help in-domain OOD generalization to hand-drawn sketches.
>
> ---
>
>
> > W3. The baseline success rates are extremely low (~10%), making “20% improvement” uninformative...
>
> We would like to clarify that fine-tuning performance is not presented as a core contribution of this paper. The fine-tuning experiments are included primarily to investigate whether it provides benefits, and if so, where those benefits occur in the Turtle Graphics domain, rather than to highlight absolute performance improvements as a key result.
>
> Regarding the data synthesis strategy, our data generation framework is introduced as a practical way to obtain training data so that we can study fine-tuning behavior. A systematic comparison with alternative data synthesis methods is beyond the scope of this benchmark-focused work and would require substantial additional experiments, which we view as valuable future work.
>
> ---
>
>
> > W4. No comparison with existing visual to program or sketch-to-code benchmarks...
>
>
> We position TURTLEAI as a domain-specific benchmark for visual-to-code reasoning in Turtle Graphics, not as a universal benchmark for all visual program-induction tasks. Existing datasets on SVG/TikZ, sketch-to-code, or UI-to-code involve different languages and evaluation setups. Building evaluation pipelines across all of them for many VLMs would require substantial additional engineering and is beyond the scope of our paper.
>
> Our setup follows a standard benchmark structure: define a well-scoped domain, introduce curated datasets, and systematically evaluate a broad set of models within that domain, thereby providing valuable insights for domain practitioners.
>
>
>
> ---
>
> > W5. The data and task space are extremely toy-like...
>
> First, despite the conceptual simplicity of the visuals, the tasks are far from trivial for current models. For example, GPT-5 only achieves 9.11% accuracy on the TURTLEAI-DS dataset, despite its strong performance on many existing multimodal benchmarks. Thus, even in this “toy-like” domain, state-of-the-art VLMs struggle substantially, indicating that TURTLEAI poses a real challenge for visual programming.
>
> Second, we frame TURTLEAI as a benchmark for visual programming in the Turtle Graphics domain. Our goal is to reveal limitations of existing multimodal models and analyze their failure modes in this setting, rather than to make claims about "general visual reasoning or broad program synthesis. We hope this clarifies your concern.

---

> > ### Author Response · Authors · 2025-11-26
> > **Rebuttal (2/2)**
> >
> > > W6. Paper presentation quality is below conference standard. All tables are mislabeled as figures, and captions are consistently placed incorrectly. This violates standard formatting guidelines and indicates insufficient care in preparation.
> >
> > We intentionally rendered all tables as figures to keep caption positions consistent across the paper. This was a formatting choice rather than an oversight, and it does not affect the clarity or correctness of the reported results. We thank the reviewer for pointing this out and will adjust the environments to follow the standard table/figure convention.
> >
> >
> > ---
> >
> > > Q1: Why are there no experiments on existing sketch-to-code or visual program induction benchmarks...
> >
> > Please see our response to W4.
> >
> > ---
> >
> > > Q2: In Fig.7, why does fine-tuning severely degrade performance on hand-drawn sketches...
> >
> >
> > Please see our response to W2.
> >
> > ---
> >
> > > Q3: What proportion of generated samples are incorrect, redundant, or semantically invalid...
> >
> > For the evaluation datasets, all samples are manually validated, so **none** of the released evaluation tasks are incorrect, redundant, or semantically invalid.
> >
> > For the training dataset, we do not perform human auditing due to its large size. However, all the samples are correct. This is because we first generate Turtle code and then execute this code to obtain the geometric drawings; only samples that execute successfully are kept.
> >
> >
> > ---
> >
> >
> > > Q4: How do you demonstrate that the proposed data synthesis approach is superior to simpler strategies (e.g., template mutation or random augmentation)?
> >
> > Please see our response to W3.
> >
> > ---
> >
> > > Q5: Many baselines used (e.g., GPT-4V, Qwen2-VL) are outdated relative to 2024–2025 VLMs. Why are recent models (GPT-4o, GPT-5，Qwen2.5-VL，Qwen3-VL，etc.) missing?
> >
> > Our benchmark already includes GPT-5 and GPT-4o in the evaluation (see Fig. 5), so these models are **not missing**. Models like Qwen2.5-VL and Qwen3-VL were released after we completed the main experiments and therefore are not included in this version. We will consider adding results for newer models in an updated version of the benchmark.
> >
> > ---
> >
> > We thank the reviewer for the feedback and hope our response clarifies the misunderstandings and addresses your concern.

---

> > > ### Comment · Reviewer_Ecws · 2025-11-27
> > > **reply to authors**
> > >
> > > Thank you for providing these detailed clarifications in your rebuttal.
> > >
> > > I have carefully read the detailed answers you provided and discovered some details that I overlooked in the appendix before.
> > > While many of my initial concerns have been addressed. And i have some follow up questions want to  discussion with you.
> > >
> > > 1.   The paper highlight the following key contributions: "Second, we propose a novel data generation framework, TURTLEAI-Datagen, that can effectively generate large-scale synthetic datasets from a small set of seed samples.      Third, we conduct comprehensive experiments and analyses on TURTLEAI, revealing valuable insights into VLMs’ capabilities and limitations."
> > >
> > > - To further demonstrate the scientific significance and potential impact of TURTLEAI-Datagen, I suggest that the authors consider conducting evaluations beyond the TURTLEAI dataset.    For example, have the authors attempted to fine-tune models using the data generated by TURTLEAI-Datagen and then evaluate them on other existing or non-TURTLEAI Turtle Graphics–related datasets (if such datasets exist)?    If strong performance can be achieved on external Turtle Graphics datasets, it would more convincingly support the framework’s generalizability and community value.
> > >
> > > - To further highlight the challenge and contemporary value of the TURTLEAI benchmark, I suggest that the authors, in addition to the already included GPT-4o and GPT-5 results, incorporate more recent vision-language models such as Qwen2.5-VL, Qwen3-VL, and Intern3-VL.    If these model evaluations can be included in the final version, it would more fully demonstrate the representativeness and comprehensiveness of the benchmark.
> > >
> > > 2.  I apologize for the oversight regarding the DSCraft data type not being included in the training set.   However, I remain concerned about the results presented in Figure 7b.   I note that the performance of the Turtle-trained models (both 7B and 72B) on the generic coding tasks (OOD test sets in Figure 7b) is significantly lower than their original base model performance, particularly for the 7B model.
> > >
> > >     Does this observed decay in general coding ability, after fine-tuning on TurtleAI-train indicate that the model is **overfitting to the narrow Turtle coding syntax**?   Could this phenomenon signal a trade-off where the model gains domain-specific expertise at the expense of its pre-existing, broader coding comprehension or generation capabilities?
> > >
> > > 3. If the effective of this data framework is not verified (that is, the validity of the proposed dataset on different models), but only the effective generation of large-scale data volume is emphasized, then does this contribution still hold true?    Of course, this includes the capabilities mentioned in the first question on different VLMS and on evaluation sets in other Turtle Graphics domains.
> > >
> > > 4. I noticed that the TURTLEAI-DS evaluation set involved human annotation.   For the larger TURTLEAI-Train, the authors mention deduplication, successful code execution, and CoT labeling performed by Qwen2-VL-72B and Pixtral-Large.  However, the LLMs are susceptible to hallucinations and inherent biases, beyond the successful execution of the final code, what additional measures or schemes were employed to ensure the correctness.
> > > Also, I have noticed that in your responses to other reviewers, my other concers have been resolved.
> > >
> > > Overall, I think you have addressed some concerns. I will consider improving my final score.

---

> > > > ### Author Response · Authors · 2025-11-28
> > > >
> > > > Thanks for the follow-up. Below we answer your questions.
> > > >
> > > > ---
> > > >
> > > > > ...consider conducting evaluations beyond the TURTLEAI dataset...
> > > >
> > > > We appreciate the suggestion. Our benchmark includes two datasets (i.e., DSBasic and DSCraft) that are **not synthetic and not used for seeding the training dataset**, which we use as *external-style* evaluations. We agree that testing on third-party Turtle Graphics datasets would further demonstrate cross-benchmark generalizability, and we see this as a promising direction for future work.
> > > >
> > > > ---
> > > >
> > > > > ...incorporate more recent vision-language models such as Qwen2.5-VL, Qwen3-VL, and Intern3-VL...
> > > >
> > > > In addition to the GPT-4o and GPT-5 results already reported in the paper, we have now evaluated the more recent vision-language models. The table below reports their symbolic success rates across our datasets:
> > > >
> > > > | | DSBasic | DSCraft | DSSyn | DS    |
> > > > | ---| --- | --- | --- | --- |
> > > > | InternVL3-78B | 12.75%  | 7.84%   | 3.23% | 4.98% |
> > > > | Qwen3-VL-30B-A3B-Instruct | 18.63%  | 13.73%  | 2.91% | 6.20% |
> > > > | Qwen2.5-VL-72B-Instruct   | 15.69%  | 17.65%  | 3.55% | 6.80% |
> > > >
> > > > These results show that even very recent large VLMs still struggle on TURTLEAI. We will incorporate these new results into the revised paper.
> > > >
> > > >
> > > > ---
> > > >
> > > > > I remain concerned about the results presented in Figure 7b... Does this observed decay in general coding ability, after fine-tuning on TurtleAI-train indicate that the model is **overfitting to the narrow Turtle coding syntax**?...
> > > >
> > > >
> > > > Thank you for raising this point. In fact, your interpretation matches our findings. As discussed in **Section 4.5 (lines 436–443)**, fine-tuning on TURTLEAI-Train improves in-domain performance but leads to a drop on out-of-domain coding benchmarks.
> > > >
> > > > Our results therefore reveal a trade-off: fine-tuning strengthens in-domain ability at the expense of some general capability, especially for smaller models. This is why we include both in-domain OOD and out-of-domain OOD experiments (**Section 4.5**).
> > > >
> > > > More broadly, this loss of generality after fine-tuning is well observed, particularly for smaller models or when many parameters are updated [1,2,3]. Prior work suggests that mixing a small fraction of general data into the fine-tuning corpus can mitigate forgetting.
> > > >
> > > > In this work, our fine-tuning experiments focus on **whether** and **why** domain-specific tuning helps on our benchmark, rather than on developing broadly general-purpose models. For this reason, we did not include mixed-domain tuning. However, we will **release all datasets** so the community can explore mixed-domain training and anti-forgetting strategies, which we see as a natural extension of our benchmark.
> > > >
> > > > ---
> > > >
> > > > > If the effective of this data framework is not verified...
> > > >
> > > > We agree that effectiveness should be shown beyond data volume. In our paper, we assess TURTLEAI-Datagen by generating training data, fine-tuning multiple models on it, and checking whether their performance improves.
> > > >
> > > > Below are some of our efforts to demonstrate the effectiveness of the proposed data framework:
> > > > - **Fine-tuning three models (Fig. 5)** We observed that all three models (Qwen2-VL-72B, Qwen2-VL-7B, Pixtral-12B) show performance gains when fine-tuned on TURTLEAI-Datagen data.
> > > > - **Studying scaling with fine-tuning data size (Fig. 6d).** We observed that finetuning performance improves steadily as the generated dataset grows.
> > > >
> > > > Since this is primarily a **benchmark** paper, our main focus is on building high-quality datasets and a systematic evaluation protocol. TURTLEAI-Datagen is included mainly to support controlled fine-tuning studies on our benchmark. We will clarify this point in the revision, and we appreciate the suggestion.
> > > >
> > > >
> > > > ---
> > > >
> > > > > ...beyond the successful execution of the final code, what additional measures or schemes were employed to ensure the correctness.
> > > >
> > > > We did not conduct additional verification of the CoT labels in TURTLEAI-Train. At this scale (738k samples), CoT is used only as auxiliary supervision for fine-tuning, **not for evaluation**. Some noise may exist, but it affects only the fine-tuned models; all evaluation sets (DSBasic, DSCraft, DSSyn) are fixed and do not rely on CoT labels.
> > > >
> > > > Empirically, CoT still helps. As shown in Fig. 7a, Pixtral-12B-TURTLE trained **with** CoT outperforms the variant trained **without** it, especially on the hand-drawn OOD dataset DSCraft, suggesting that CoT labels improves generalization even with potential noisy labels.
> > > >
> > > > ---
> > > >
> > > > References:
> > > >
> > > > [1] AgentTuning: Enabling Generalized Agent Abilities for LLMs. ACL (Findings) 2024
> > > >
> > > > [2] Revisiting Catastrophic Forgetting in Large Language Model Tuning. EMNLP (Findings) 2024
> > > >
> > > > [3] Recall and Learn: Fine-tuning Deep Pretrained Language Models with Less Forgetting. EMNLP 2020
> > > >
> > > > ---
> > > >
> > > > We hope these responses clarify our contributions and address your concerns. Thank you for the thoughtful feedback. We will revise the paper accordingly.

---

### Official Review · Reviewer_wESH · 2025-11-01

**Soundness:** 3
**Presentation:** 3
**Contribution:** 2
**Rating:** 2
**Confidence:** 5

**Summary:**

The paper introduces TURTLEAI, a multimodal benchmark designed to assess visual programming and reasoning in Turtle Graphics. It provides 823 tasks requiring models to generate Python code that reproduces target images. Experiments on 20 VLMs show that state-of-the-art models struggle, with GPT-4o and Qwen2-VL-72B achieving low success rates. The authors also propose TURTLEAI-Datagen, a synthetic data generation pipeline producing 700k+ samples from 10 seeds, improving model performance by over 20% after fine-tuning.

**Strengths:**

1. The paper offers a well-structured task setup linking visual reasoning to program synthesis and conducts systematic experiments across many VLMs, yielding a solid empirical assessment of model limitations in structured visual-to-code settings.

2. The synthetic data framework is executed at scale and empirically improves model performance, demonstrating practical value for enhancing VLM capability in controlled visual programming tasks.

**Weaknesses:**

1. Limited novelty relative to prior benchmarks: Similar multimodal visual-to-code and graphics reasoning benchmarks already exist (e.g., NAACL 2025 TurtleBench: A Visual Programming Benchmark in Turtle Geometry https://aclanthology.org/2025.naacl-long.607/
). The paper primarily repackages an existing paradigm (image-to-code in a constrained graphics domain) rather than introducing a fundamentally new task or evaluation angle.

2. Narrow and arguably low-impact domain: Turtle Graphics is a highly simplified, pedagogical environment with limited real-world relevance. Performance in this synthetic sandbox does not clearly translate to practical multimodal programming, robotics, CAD/graphics reasoning, or general visual planning tasks. The paper lacks evidence that gains on Turtle tasks meaningfully correlate with improvements on broader multimodal program synthesis or vision-reasoning benchmarks, raising questions about external validity and actual scientific payoff.

3. Evaluation and insights remain shallow: While the paper reports success rates and shows synthetic data improves scores, the analysis stops short of deeper failure categorization, ablation across visual complexity factors, or diagnostics that could reveal why models fail (e.g., perceptual ambiguity vs. planning vs. code correctness). The work also does not benchmark against alternative data augmentation or curriculum approaches, nor does it explore whether improvements generalize beyond the Turtle setting, limiting interpretability and impact of the proposed method.

**Questions:**

N/A

---

> ### Author Response · Authors · 2025-11-26
>
> Dear Reviewer,
>
> Thank you for your thoughtful comments and feedback. We will address your concerns as follows.
>
> ---
>
> > W1. Limited novelty relative to prior benchmarks...
>
> We agree that our benchmark and TurtleBench belong to the same turtle-graphics family, but our benchmark differs in task design and evaluation. Below we compare TURTLEAI with TurtleBench from these two perspectives.
>
> 1. Task perspective
>     - **More tasks and richer structure.** TurtleBench has 260 tasks. TURTLEAI includes 823 evaluation tasks spanning six geometric categories and three difficulty levels, while TurtleBench contains only two task types (Scratch vs. Tweak).
>     - **More diverse and challenging tasks.** TurtleBench focuses on black-and-white abstract geometric shapes built from a small set of operations. TURTLEAI additionally introduces composite tasks that require combining multiple transformations and colorful patterns where models must correctly match both structure and color, which significantly increases difficulty.
>     - **Real-world and hand-drawn tasks.** Our benchmark contains tasks (DSBasic) actively used by real students on the XLogoOnline educational platform. TURTLEAI also provides a hand-drawn variant, enabling testing under out-of-distribution visual styles. TurtleBench uses only manual tasks and does not include a hand-drawn subset.
> 2. Model evaluation perspective
>     - **Broader model coverage and fine-tuning.** TurtleBench evaluates 3 models. TURTLEAI evaluates 20 models and includes fine-tuned variants, allowing us to analyze both absolute capability and how fine-tuning impacts specific failure modes.
>
> Below is the table comparing the two benchmarks.
>
> |       | TURTLEAI (Ours)     | TurtleBench     |
> | --- | ---| ----|
> | # Tasks           | 823 tasks| 260 tasks    |
> | Task Taxonomy     | 6 geometric categories | 2 task types   |
> | Difficulty Levels | Easy / Medium / Hard  | No difficulty levels  |
> | Task Diversity    | Includes composite tasks and colorful patterns  | Black-and-white abstract shapes    |
> | Datasets          | DSBasic (tasks used by real students on XLogoOnline); DSCraft (hand-drawn tasks); DSSyn (synthetic tasks) | All tasks are manually generated ; no real-world source |
> | Evaluated Models  | 20 VLMs evaluated (including fine-tuned models) | 3 models evaluated, no fine-tuned models  |
>
> ---
>
> > W2. Narrow and arguably low-impact domain...
>
> In short, our benchmark is useful both for tracking multimodal model performance and for supporting real use cases in educational settings.
>
> First, we agree that Turtle Graphics is a simplified, pedagogical setting, and we do not claim that good performance here automatically transfers to robotics, CAD, or general visual planning tasks. Our goal is different: we use this domain to diagnose what multimodal models can and cannot do. For instance, we found that when the target shape is a square with a specific cut-off, the model often just draws a normal square and ignores the cut. These are not random mistakes; they show a consistent tendency to default to common shapes instead of respecting fine-grained geometry. Our benchmark makes these errors immediately visible and measurable.
>
> Second, we believe the domain is more impactful than it might appear, especially in the educational context. Turtle-style environments are widely used in K–12 and introductory programming courses to teach basic coding, geometry, and computational thinking (e.g., XLogoOnline, Oxford Turtle System, Turtle360, and Turtle Academy). If we want to use multimodal models as coding tutors or hint/feedback generators in these settings, they must handle such tasks reliably. Our results show that current models still struggle with these "simple but precise" tasks, revealing an important limitation in fine-grained visual reasoning that matters both scientifically and for real educational applications.
>
> We hope this clarifies our motivation and the relevance of the domain.
>
> ---
>
> > W3. Evaluation and insights remain shallow....
>
> We respectfully disagree that our analysis stops short of deeper failure categorization or ablations; these analyses are already included in the paper:
> - **Failure categorization and diagnostics.** We manually categorize DSBasic errors into seven types (visual, decomposition, spatial, programming, precision, repetition, evaluation) and analyze their distribution in **Figure 6**.
> - **Ablation across visual complexity factors.** We compare performance by task category, difficulty level, and dataset (**Figure 12**).
>
> Our work primarily focuses on benchmarking existing multimodal models, rather than comparing data augmentation or curriculum learning techniques. We agree that exploring such approaches is an interesting direction; however, it falls outside the scope of this paper, and we leave it for future work.
>
> ---
>
> We hope this response addresses the reviewer’s concerns. We will also incorporate the reviewer’s valuable feedback into the revised version of our paper.

---

### Note · Authors · 2025-12-29

I have read and agree with the venue's withdrawal policy on behalf of myself and my co-authors.